# KooNPro: A Variance-Aware Koopman Probabilistic Model Enhanced by Neural Process for Time Series Forecasting

**Ronghua Zheng, Hanru Bai & Weiyang Ding** [*]
Institute of Science and Technology for Brain-inspired Intelligence
Fudan University, Shanghai 200433, China
rhzheng23@m.fudan.edu.cn, hrbai23@m.fudan.edu.cn, dingwy@fudan.edu.cn

## Abstract

The probabilistic forecasting of time series is a well-recognized challenge, particularly in disentangling correlations among interacting time series and addressing the complexities of distribution modeling. By treating time series as temporal dynamics, we introduce **KooNPro**, a novel probabilistic time series forecasting model that combines variance-aware deep **Koo**pman model with **N**eural **Pro**cess. KooNPro introduces a variance-aware continuous spectrum using Gaussian distributions to capture complex temporal dynamics with improved stability. It further integrates the Neural Process to capture fine dynamics, enabling enhanced dynamics capture and prediction. Extensive experiments on nine real-world datasets demonstrate that KooNPro consistently outperforms state-of-the-art baselines. Ablation studies highlight the importance of the Neural Process component and explore the impact of key hyperparameters. Overall, KooNPro presents a promising novel approach for probabilistic time series forecasting. Code is available at this repository: https://github.com/Rrh-Zheng/Koonpro.

## 1 Introduction

Time series forecasting is essential in supply chain management, finance, energy, and healthcare. Probabilistic forecasting quantifies uncertainty, supporting robust decision-making in tasks like inventory optimization and risk management. For instance, in supply chains, it helps businesses anticipate demand fluctuations, optimize planning, and improve efficiency. Traditional methods like Auto-Regressive Integrated Moving Average (ARIMA) (Said & Dickey, 1984) and the Kalman filter family (Auger et al., 2013) struggle with the complexities of real-world time series. Deep learning approaches for time series forecasting have evolved from Recurrent Neural Networks (RNNs) (Salehinejad et al., 2017) to Long Short-Term Memory (LSTM) networks (Yu et al., 2019). The introduction of the attention mechanism (Vaswani et al., 2017) further advanced the field, leading to Transformer-based models such as Zhou et al. (2021), Wu et al. (2021), and Zhou et al. (2022). More recently, diffusion probabilistic models (Ho et al., 2020; Rasul et al., 2021; Li et al., 2022; Fan et al., 2024) have emerged as a promising paradigm for probabilistic forecasting, with works like Kollovieh et al. (2024) demonstrating their potential. Moreover, foundation models for time series forecasting, such as Ansari et al. (2024), have introduced the concept of learning a universal representation for diverse time series tasks. Despite these advancements, many deep learning models focus primarily on point estimation, which limits their ability to fully capture the uncertainty and dynamics inherent in complex systems. State space models, such as Rangapuram et al. (2018a), Wang et al. (2019) and Paria et al. (2021) offer an alternative by modeling time series as low-dimensional latent decompositions. These methods excel at representing temporal dynamics in reduced spaces but often rely on fixed assumptions about state transitions, which may fail to generalize to highly nonlinear and non-stationary scenarios.

Dynamic Mode Decomposition (DMD) is a powerful tool for modeling time series as dynamic systems (Kuttichira et al., 2017), aiming to identify the underlying dynamics and subsequently

---

[*]Corresponding author

utilize them for prediction (Kou & Zhang, 2019; Yuan et al., 2021). However, DMD's reliance on discrete eigenvalues (point-spectra) restricts its ability to describe the nonlinear patterns of complex systems. The Koopman theory offers a powerful framework for analyzing nonlinear systems by representing them in a linear but infinite-dimensional space. The ability to decompose complex dynamics into Koopman eigenfunctions has inspired many works to incorporate this theory into time series modeling. Combining the Koopman theory (Koopman, 1931) with deep learning (Lusch et al., 2018) enhances the ability to model temporal dynamics with greater sophistication, leading to more accurate predictions. For example, models such as Liu et al. (2023) and Wang et al. (2023) enhance predictive capabilities through deep Koopman theory by learning the continuous-spectra of dynamics. However, these models are often affected by spectrum pollution, leading to unstable convergence and reduced accuracy, especially in highly nonlinear systems (Colbrook & Townsend, 2024). To address this, the concept of pseudo-spectra has been introduced. Pseudo-spectra accounts for the latent spaces where eigenvalues may appear under different perturbation conditions, which makes them more robust to noise and better suited for capturing continuous variations.

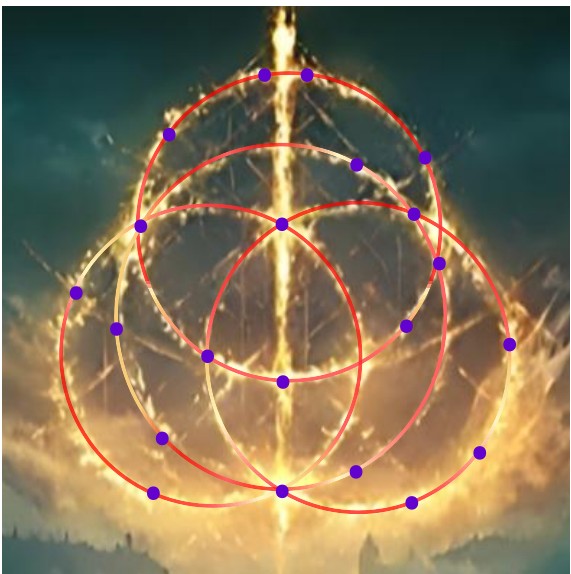

Figure 1: To enhance the accuracy and robustness of probabilistic forecasting, we model time series as dynamic evolutions in a hidden space. In our setting, the dynamic system is constrained within unit circles. The point-spectra (purple), continuous-spectra (red), and pseudo-spectra (yellow) are represented within unit circles. Multiple unit circles can be deemed as different time points in the hidden space. Figure adapted from `Elden Ring` (© FromSoftware Inc. and Bandai Namco Entertainment Inc.), modified and used under fair use for academic purposes.

The motivation of this work is to assemble various spectral components within a unified framework, as illustrated in Fig. 1. **Point-spectra** are incorporated via Neural Process (Garnelo et al., 2018b) to capture global temporal dynamics. **Continuou-spectra** are modeled using the Koopman theory to represent local temporal dynamics. For **Pseudo-spectra**, we propose a variance-aware continuous-spectra modeled by Gaussian distributions, drawing on the idea that probability distributions can characterize pseudo-spectra within a neighborhood by quantifying overall dispersion (Colbrook et al., 2024).

By modeling time series as dynamic systems grounded in multi-spectra representations, we propose KooNPro, a novel probabilistic framework that combines global latent features from point-spectra via Neural Process with variance-aware modeling of continuous-spectra. We adopt variational inference and derive a tailored Evidence Lower Bound (ELBO) to optimize the unified model. KooNPro achieves high predictive accuracy and robustness in high-dimensional, non-stationary settings, demonstrating state-of-the-art performance. Our contributions are summarized as follows:

- We introduce KooNPro, a novel probabilistic prediction model that synergistically integrates the probabilistic Koopman model with Neural Process. Drawing inspiration from perturba-

tions in pseudo-spectra, KooNPro incorporates variance-aware continuous-spectra to learn local temporal dynamics effectively. Additionally, it utilizes Neural Process to capture the full point-spectra which represented the global temporal dynamics, facilitating improved predictive capabilities.

- Extensive experiments conducted on diverse real-world datasets convincingly demonstrate the superiority of our proposed model. It consistently achieves state-of-the-art performance, significantly outperforming existing methods across two metrics.

- We perform comprehensive ablation studies demonstrating the ability of Neural Process to capture temporal dynamics globally, thereby enhancing prediction performance. We also assess the impact of key hyperparameters on model efficacy. Additionally, we provide a detailed case study that visually illustrates the prediction performance of KooNPro, yielding intuitive and interpretable results.

## 2    RELATED WORK

This work engages with three key areas: probabilistic time series prediction, the probabilistic Koopman model, and Neural Process. While each of these fields has received considerable attention, we limit our discussion to the most pertinent studies to ensure brevity. A more extensive version can be found in Appendix A.

**Probabilistic time series prediction:** Recent developments in probabilistic forecasting for time series integrate deep learning, statistical approaches, and diffusion models. Rangapuram et al. (2018b), Salinas et al. (2020) and Li et al. (2021) combine state space model with deep learning. Feng et al. (2023) and Tang & Matteson (2021) developed attention-based mechanism that enhance long-range dependencies for improved forecast accuracy. Gaussian Process (GP), combined with temporal decomposition, was used by Yan et al. (2021), Nguyen & Quanz (2021), and Salinas et al. (2019) to better model uncertainty in multivariate settings. The data distribution-based generative model proposed by (Gouttes et al., 2021) and the Kalman filter-based approach proposed by de Bézenac et al. (2020), blend probabilistic techniques for better scalability and robustness. Diffusion models, such as those proposed by Rasul et al. (2021), Li et al. (2022), and Fan et al. (2024), model forecasting as a denoising task, excelling in high-dimensional settings.

**Probabilistic Koopman model:** The Koopman theory has been developed in lots of fields, the most related work is the probabilistic Koopman model, originally proposed by Morton et al. (2019). Han et al. (2022) designed a stochastic Koopman neural network for control, in which the latent observables are represented through Gaussian distribution. Colbrook et al. (2024) integrates the concept of variance into the Koopman framework by introducing variance-aware pseudo-spectra, thereby ensuring convergence within the model. For time series tasks, Naiman et al. (2023) utilized Koopman theory to represent the latent conditional prior dynamics via a linear map, while Mallen et al. (2024) introduced a framework enabling probabilistic forecasting for systems with periodically varying uncertainty.

**Neural process:** Neural Process (NP), first proposed by Garnelo et al. (2018a) as Conditional Neural Process, bridges neural networks' scalability and Gaussian Process's ability to model uncertainty. Garnelo et al. (2018b) provide flexible, probabilistic function approximations, making them efficient and adaptable across tasks. Attentive Neural Process (Kim et al., 2019), Convolutional Conditional Neural Process (Gordon et al., 2019b), Gaussian Neural Process (Bruinsma et al., 2021) and Autoregressive Neural Process (Bruinsma et al., 2023) deploy different methods to probe the relation between input-output pairs. Other works extend NP to meta-learning, such as Meta-Learning Stationary Stochastic Process (Foong et al., 2020) and Meta-Learning Probabilistic Inference (Gordon et al., 2019a). A comprehensive survey by Jha et al. (2022) has organized these developments, exploring their wide applications in uncertainty-aware learning.

## 3    BACKGROUND

Due to space constraints, we discuss the background details in Appendix B and focus here on the core of Koopman theory and Neural Process, which form the foundation of KooNPro.

The spectrum decomposition of the Koopman operator is crucial for understanding dynamics. The point-spectra reveal modal contributions via eigenvalues and corresponding eigenfunctions, which can reveal how different modes contribute to the system's evolution. However, many complex systems exhibit continuous-spectra, reflecting chaotic or highly irregular behavior. In such cases, the Koopman operator's spectrum forms a continuum rather than isolated eigenvalues and offers a sophisticated framework for capturing both regular and chaotic components of system dynamics. Although the Koopman operator is effective for predictive modeling but is prone to spectrum pollution, which can hinder convergence and stability. Drawing inspiration from pseudo-spectra perturbations and the idea that probability distributions capture variable dispersion, we propose variance-aware continuous-spectra modeled by Gaussian distributions.

Neural Process (NP) is a stochastic process $p(f)$ that describes the predictive distribution over the target set $(\boldsymbol{x}_D, \boldsymbol{y}_D) := (\boldsymbol{x}_i, \boldsymbol{y}_i)_{i \in D}$ given the context set $(\boldsymbol{x}_C, \boldsymbol{y}_C) := (\boldsymbol{x}_i, \boldsymbol{y}_i)_{i \in C}$. Garnelo et al. (2018b) prompts use the distribution of a high-dimensional random vector $\boldsymbol{S}$ to represent $p(f)$ as

$$p(\boldsymbol{y}_D | \boldsymbol{x}_D, \boldsymbol{x}_C, \boldsymbol{y}_C) := \int p(\boldsymbol{y}_D | \boldsymbol{x}_D, \boldsymbol{S}) p(\boldsymbol{S} | \boldsymbol{x}_C, \boldsymbol{y}_C) d\boldsymbol{S}. \tag{1}$$

NP consists of two components: an encoder that maps the input-output pairs of the context set $(\boldsymbol{x}_C, \boldsymbol{y}_C)$ to $\boldsymbol{S}$ for representing $p(f)$, and a decoder that combines $\boldsymbol{S}$ with $\boldsymbol{x}_D$ to generate $\boldsymbol{y}_D$. Due to the intractable log-likelihood, NP adopts amortized variational inference and maximizes the evidence lower bound (ELBO) of the log-likelihood as follows

$$\log(\boldsymbol{y}_D | \boldsymbol{x}_D, \boldsymbol{y}_C, \boldsymbol{x}_C) \geq \mathbb{E}_{\boldsymbol{S} \sim q(\boldsymbol{S} | \boldsymbol{x}_D, \boldsymbol{y}_D)} \left[ \log p(\boldsymbol{y}_D | \boldsymbol{x}_D, \boldsymbol{S}) \right] - D_{KL} \left( q(\boldsymbol{S} | \boldsymbol{x}_D, \boldsymbol{y}_D) \| p(\boldsymbol{S} | \boldsymbol{x}_C, \boldsymbol{y}_C) \right). \tag{2}$$

## 4 METHOD

This section offers a detailed overview of KooNPro, with the complete architecture illustrated in Fig.2. KooNPro leverages the synergistic integration of Neural Process (NP) with a probabilistic deep Koopman model to learn temporal dynamics for probabilistic future prediction. Initially, NP globally captures the point-spectra of temporal dynamics to govern the full time series, as indicated by the downward arrows in Fig.2. Additionally, inspired by the concept of pseudo-spectra, we utilize a probabilistic deep Koopman model to refine continuous-spectra, yielding variance-aware spectra for a more nuanced representation of local temporal dynamics, as demonstrated by the shadowed box in Fig.2.

### 4.1 CAPTURE TEMPORAL DYNAMICS BY NEURAL PROCESS

Dynamic Mode Decomposition (DMD) is a classical method that elucidates the relationship between time series data and corresponding dynamics. However, DMD's reliance on point-spectra limits its capacity to capture the nonlinear patterns inherent in complex systems, thereby constraining its predictive power. To address these limitations, KooNPro integrates Neural Process (NP) with DMD to generate a stochastic process that captures the global dynamics of the whole time series more effectively. Specifically, KooNPro identifies the distribution of the latent variable $\boldsymbol{S} \in \mathbb{R}^s$, which integrates the underlying dynamics of a time series in Eq.1, as shown in the left part of Fig.2.

We utilize Takens' theory like Yuan et al. (2021), define a time series $\boldsymbol{z}_{1:T} \in \mathbb{R}^{T \times d \times k}$, where $T$ denotes the time length, $d$ denotes the feature dimension and $k$ denotes the delay-embedded length, and let $\boldsymbol{x} = \boldsymbol{z}_{1:T-1}$, $\boldsymbol{y} = \boldsymbol{z}_{2:T}$. To estimate the dynamics governing time series in the context set, KooNPro employs a neural network $\boldsymbol{\tau}$ to embed $\hat{\boldsymbol{x}} \in \mathbb{R}^{(T-1) \times d \times 1}$ and $\hat{\boldsymbol{y}} \in \mathbb{R}^{(T-1) \times d \times 1}$ as the initial components of delay embedding like

$$\tau_C := \boldsymbol{\tau}(\boldsymbol{x}_C, \boldsymbol{y}_C) = \boldsymbol{\psi} \left( \frac{1}{c} \sum_{i=1}^{c} vec\left( \hat{\boldsymbol{x}}^\dagger \hat{\boldsymbol{y}} \right) \right), \tag{3}$$

where $\boldsymbol{\psi}$ denotes a learnable Multi-Layer Perceptron (MLP), $c$ denotes the number of items in the context set, and the eigenvalues of $\hat{\boldsymbol{x}}^\dagger \hat{\boldsymbol{y}}$ can approximate the point-spectra of the corresponding time series dynamics. We model the distribution of $\boldsymbol{S}$ by a factorized Gaussian distribution parametrized by $\tau_C$, i.e.,

$$p(\boldsymbol{S} | \boldsymbol{x}_C, \boldsymbol{y}_C) = \mathcal{N}(\boldsymbol{S}; \mu(\tau_C), \sigma(\tau_C)). \tag{4}$$

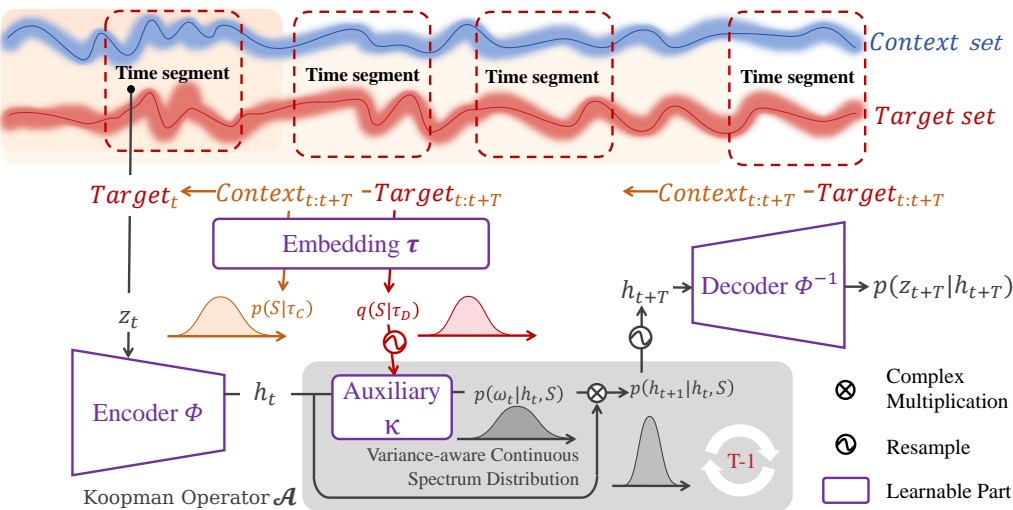

Figure 2: **The complete architecture of KooNPro:** $p(S|\tau_D)$ and $p(S|\tau_C)$ are the latent representations of DMD mean over the target set and the context set respectively. Given the first time step $z_1$ in the target set, the encoder $\phi$ generates the latent variable $h_1$. We apply the Koopman operator $\mathcal{A}$ to $h_1$ for $T - 1$ steps then generate the distribution of $h_{2:T}$. Finally, the decoder $\phi^{-1}$ maps $h_{2:T}$ back to the original space to generate the distribution of $z_{2:T}$.

Denote $p(S|\tau_C) := p(S|x_C, y_C)$ for any set $C$, thus $p(S)$ represents the distribution of underlying dynamics according to different time series sets. The ELBO in Eq.2 can be reformulated as follows

$$\log(\boldsymbol{y}_D|\boldsymbol{x}_D, \boldsymbol{y}_C, \boldsymbol{x}_C) \geq \mathbb{E}_{\boldsymbol{S} \sim q(\boldsymbol{S}|\tau_D)}[\log p(\boldsymbol{y}_D|\boldsymbol{x}_D, \boldsymbol{S})] - D_{KL}(q(\boldsymbol{S}|\tau_D)||p(\boldsymbol{S}|\tau_C)). \tag{5}$$

The second term on the right of Eq.5 captures the difference in the distribution of $S$, reflecting the diversity of temporal dynamics across different time series. Although we employ $\tau$ to embed the point-spectra that characterize the global temporal dynamics, it provides only a coarse approximation. For the first term on the right side of Eq.5, we maximize it using a probabilistic deep Koopman model, which provides a more nuanced representation of the local temporal dynamics through the lens of the pseudo-spectra.

## 4.2 PROBABILISTIC DEEP KOOPMAN MODEL

The proposed probabilistic deep Koopman model shown in the shadowed box of Fig.2 concentrates on explaining local characteristics of time series with variance-aware continuous-spectra. This approach offers a deeper understanding of temporal dynamics behavior over time, leading to improved predictive capabilities. In essence, estimating the true temporal dynamics is equivalent to maximizing the expectation $\mathbb{E}_{q(\boldsymbol{S}|\tau_D)}[\log p(\boldsymbol{y}_D|\boldsymbol{x}_D, \boldsymbol{S})]$.

For simplicity, we discuss only the data in the target set and omit the index $D$. Following Lusch et al. (2018), we compose KooNPro by three MLPs: an **encoder** $\phi : \mathbb{R}^{d \times k} \to \mathbb{R}^n$ and a **decoder** $\phi^{-1} : \mathbb{R}^n \to \mathbb{R}^{d \times k}$ to identify an appropriate linear space, along with an **auxiliary** $\kappa : \mathbb{R}^{n+s} \to \mathbb{R}^n$ to learn the continuous-spectra of the corresponding temporal dynamics in linear space. Firstly, KooNPro vectorized the delay-embedded time series $\boldsymbol{x} = \boldsymbol{z}_{1:T-1}$ from size $(T - 1) \times d \times k$ to $(T - 1) \times (d \cdot k) \times 1$, then applies the encoder as $\boldsymbol{h}_{1:T-1} = \phi(\boldsymbol{x})$. We hypothesize the latent space created by $\phi$ possesses linear characteristics, thus the dynamics in such space can be described as follows

$$\boldsymbol{h}_{t+1} = e^{\boldsymbol{\lambda}_t \Delta t} \boldsymbol{h}_t. \tag{6}$$

We assume $\boldsymbol{h}_t$ represents the eigenvalue of the dynamics, consisting of a pair of conjugate numbers, and $\boldsymbol{\lambda}_t$ to be pure imaginary like Lange et al. (2021), let $\boldsymbol{\lambda}_t = j\boldsymbol{\omega}_t$, thus we have

$$\mathbf{Re}(\boldsymbol{h}_{t+1}) = \mathbf{Re}(\boldsymbol{h}_t) \odot \cos(\boldsymbol{\omega}_t \Delta t) - \mathbf{Im}(\boldsymbol{h}_t) \odot \sin(\boldsymbol{\omega}_t \Delta t), \tag{7}$$

$$\mathbf{Im}(\boldsymbol{h}_{t+1}) = \mathbf{Re}(\boldsymbol{h}_t) \odot \sin(\boldsymbol{\omega}_t \Delta t) + \mathbf{Im}(\boldsymbol{h}_t) \odot \cos(\boldsymbol{\omega}_t \Delta t), \tag{8}$$

where $\odot$ is the element-wise product. We can simplify Eq.7 and Eq.8 to

$$\boldsymbol{h}_{t+1} := \mathcal{A}(\boldsymbol{h}_t, \boldsymbol{\omega}_t). \tag{9}$$

In order to learn the Koopman operator $\mathcal{A}$, which is conditioned on the global temporal dynamics representation $\boldsymbol{S}$, KooNPro utilizes the auxiliary $\boldsymbol{\kappa}$ to project the concatenation of $\boldsymbol{h}_t$ and $\boldsymbol{S}$ to $\kappa_t$,

$$\boldsymbol{\kappa} : \begin{bmatrix} \boldsymbol{h}_t \\ \boldsymbol{S} \end{bmatrix} \mapsto \kappa_t. \tag{10}$$

Drawing on the theory of pseudo-spectra, KooNPro aims to learn dynamic information not only from the eigenvalues but also from their neighborhoods. To facilitate this, KooNPro employs the auxiliary network to generate the corresponding Gaussian distribution of $\boldsymbol{\omega}_t$ parameterized by $\kappa_t$ as

$$p(\boldsymbol{\omega}_t | \boldsymbol{h}_t, \boldsymbol{S}) = \mathcal{N}(\boldsymbol{\omega}_t; \mu(\kappa_t), \sigma(\kappa_t)). \tag{11}$$

To simplify KooNPro, we assume the independence of $\boldsymbol{h}_{1:T-1}$ on the linear space. Consequently, we can describe the uncertain dynamics as

$$p(\boldsymbol{\omega}_{1:T-1} | \boldsymbol{h}_{1:T-1}, \boldsymbol{S}) = \prod_{t=1}^{T-1} \mathcal{N}(\boldsymbol{\omega}_t; \mu(\kappa_t), \sigma(\kappa_t)). \tag{12}$$

Since $\boldsymbol{h}_t = \mathcal{A}(\boldsymbol{h}_{t-1}, \boldsymbol{\omega}_{t-1})$, it yields that $p(\boldsymbol{h}_T | \boldsymbol{\omega}_{1:T-1}, \boldsymbol{h}_{1:T-1}, \boldsymbol{S}) = 1$. Thus, the variance-aware continuous-spectra in the linear space can be characterized within the framework of the Gaussian distribution like

$$p(\boldsymbol{h}_{2:T} | \boldsymbol{h}_{1:T-1}, \boldsymbol{S}) = p(\boldsymbol{h}_T | \boldsymbol{\omega}_{1:T-1}, \boldsymbol{h}_{1:T-1}, \boldsymbol{S}) p(\boldsymbol{\omega}_{1:T-1} | \boldsymbol{h}_{1:T-1}, \boldsymbol{S}) \tag{13}$$

$$= \prod_{t=1}^{T-1} \mathcal{N}(\boldsymbol{\omega}_t; \mu(\kappa_t), \sigma(\kappa_t)). \tag{14}$$

Utilizing the decoder $\phi^{-1}$, KooNPro maps the temporal dynamic evolution from the linear space back to the original space, as follows

$$p(\boldsymbol{z}_{2:T} | \boldsymbol{h}_{2:T}) = \prod_{t=2}^{T} \mathcal{N}(\boldsymbol{z}_t; \mu(\phi^{-1}(\boldsymbol{h}_t)), \sigma(\phi^{-1}(\boldsymbol{h}_t))). \tag{15}$$

The likelihood $p(\boldsymbol{y}|\boldsymbol{x}, \boldsymbol{S})$ in Eq.5, where $\boldsymbol{x} = \boldsymbol{z}_{1:T-1}$ and $\boldsymbol{y} = \boldsymbol{z}_{2:T}$, can be derived as

$$p(\boldsymbol{y}|\boldsymbol{x}, \boldsymbol{S}) = p(\boldsymbol{z}_{2:T} | \boldsymbol{h}_{1:T-1}, \boldsymbol{S}) = p(\boldsymbol{z}_{2:T} | \boldsymbol{h}_{2:T}) p(\boldsymbol{h}_{2:T} | \boldsymbol{h}_{1:T-1}, \boldsymbol{S}), \tag{16}$$

given that the encoder $\phi$ is a deterministic function. Insert Eq.14 and Eq.15 to Eq.16, we can draw the expectation as

$$\mathbb{E}_{\boldsymbol{S} \sim q(\boldsymbol{S}|\tau_D)}[\log p(\boldsymbol{y}_D | \boldsymbol{x}_D, \boldsymbol{S})] = \mathbb{E}_{\boldsymbol{S} \sim q(\boldsymbol{S}|\tau_D)}[\sum_{t=2}^{T} \log(p(\boldsymbol{z}_t | \boldsymbol{h}_t)) + \sum_{t=1}^{T-1} \log(p(\boldsymbol{h}_{t+1} | \boldsymbol{h}_t, \boldsymbol{S}))]. \tag{17}$$

The first term can be interpreted as the prediction loss, while the second term represents the linear loss, as described in Lusch et al. (2018). By maximizing the ELBO in Eq.5, we derive that $\boldsymbol{S}_C$ approximates the dynamics for each time series present during the training phase. Consequently, our predictions in the test phase commence with the initial $\boldsymbol{z}_1 \in \mathbb{R}^{1 \times d \times k}$ like

$$\boldsymbol{z}_{\tilde{T}} = \phi^{-1}(\mathcal{A}^{\tilde{T}-1}(\phi(\boldsymbol{z}_1), \boldsymbol{S}_C)). \tag{18}$$

It is crucial to note that the time length $T$ during the training stage is distinct from the prediction horizon $\tilde{T}$, with the input history length solely determined by the delay-embedded length $k$. This enables us to train KooNPro once and then forecast future values of arbitrary length.

## 5 EXPERIMENTS

In this section, we conduct comprehensive experiments on nine real-world datasets to evaluate the performance of KooNPro against state-of-the-art baselines. We demonstrate Neural Process's ability to enhance KooNPro's predictive performance and investigate the impact of key hyperparameters through ablation studies. Finally, we visualize the prediction results for the `Solar` dataset and analyze the relationship between prediction error and variance.

## 5.1 SETTINGS

**Datasets.** We consider nine real-world datasets characterized by a range of temporal dynamics, namely `ETTs`, `Solar`, `Electricity`, `Traffic`, `Taxi`, and `KDD-cup`. The data is recorded at intervals of 15 minutes, 30 minutes, 1 hour, or 1 day frequencies. Refer to Appendix C for details. All datasets are split chronologically and adopt the same train/validation/test ratios, *i.e.*, 7:1:2.

**Evaluation Metrics.** Following previous work (Fan et al., 2024), we assess our model and all baselines using $\mathbf{CRPS_{sum}}$ (Continuous Ranked Probability Score), a widely used metric for probabilistic time series forecasting, as well as $\mathbf{NRMSE_{sum}}$ (Normalized Root Mean Squared Error). The details of metrics are shown in Appendix D.

**Baselines.** We assess the predictive performance of KooNPro in comparison with multivariate time series forecasting models, including GP-Copula (Salinas et al., 2019), Transformer-MAF (Rasul et al., 2021), TimeGrad (Rasul et al., 2021), TACTiS (Ashok et al., 2023), $D^3$VAE (Li et al., 2023), DPK (Mallen et al., 2024), and MG-TSD (Fan et al., 2024). The more details of baselines can be found in Appendix E.

**Implementation details.** The training process is early stopped within 5 epochs using the Adam optimizer with a fixed learning rate of $10^{-5}$. The mini-batch size is set to 128. Additional hyperparameters, such as time length $T$, delay-embedded length $k$, and layers of MLPs are detailed in Appendix F. All models are trained and tested on a single NVIDIA RTX4070Ti 12GB GPU.

## 5.2 RESULTS

In this section, we compare the performance of KooNPro with baselines using $\mathbf{CRPS_{sum}}$ and $\mathbf{NRMSE_{sum}}$. More comprehensive results, including long-term predictions (Appendix G), performance in different noisy scenarios (Appendix H), and the relationship between performance and memory consumption (Appendix I), can be found in the appendix.

Table 1: Comparison of $\mathbf{CRPS_{sum}}$ (denoted as C-s, smaller is better) and $\mathbf{NRMSE_{sum}}$ (denoted as N-s, smaller is better) across nine real-world datasets. The means and standard errors are based on 10 independent runs of retraining and evaluation. The best performances are highlighted in red and the second are in blue.

| Model | Metric | ETTh1 | ETTh2 | ETTm1 | ETTm2 | Solar | Electricity | Traffic | Taxi | Cup |
|---|---|---|---|---|---|---|---|---|---|---|
| GP-Copula | C-s | $0.537_{\pm0.019}$ | $0.264_{\pm0.023}$ | $0.241_{\pm0.043}$ | $0.147_{\pm0.019}$ | $0.305_{\pm0.024}$ | $0.078_{\pm0.035}$ | $0.199_{\pm0.008}$ | $0.286_{\pm0.066}$ | $0.217_{\pm0.000}$ |
| | N-s | $0.835_{\pm0.032}$ | $0.383_{\pm0.032}$ | $0.424_{\pm0.088}$ | $0.172_{\pm0.027}$ | $0.671_{\pm0.034}$ | $0.122_{\pm0.058}$ | $0.293_{\pm0.018}$ | $0.412_{\pm0.115}$ | $0.346_{\pm0.010}$ |
| Trans-MAF | C-s | $0.800_{\pm0.049}$ | $0.223_{\pm0.012}$ | $0.379_{\pm0.029}$ | $0.228_{\pm0.040}$ | $0.964_{\pm0.013}$ | $0.144_{\pm0.046}$ | $0.477_{\pm0.021}$ | $0.403_{\pm0.046}$ | $0.257_{\pm0.013}$ |
| | N-s | $1.285_{\pm0.188}$ | $0.315_{\pm0.024}$ | $0.577_{\pm0.082}$ | $0.356_{\pm0.065}$ | $1.665_{\pm0.022}$ | $0.245_{\pm0.079}$ | $0.762_{\pm0.031}$ | $0.598_{\pm0.029}$ | $0.390_{\pm0.010}$ |
| Timegrid | C-s | $0.547_{\pm0.022}$ | $0.241_{\pm0.002}$ | $0.227_{\pm0.029}$ | $0.212_{\pm0.015}$ | $0.594_{\pm0.011}$ | $0.044_{\pm0.010}$ | $0.455_{\pm0.020}$ | $0.327_{\pm0.058}$ | $0.271_{\pm0.087}$ |
| | N-s | $0.889_{\pm0.039}$ | $0.325_{\pm0.000}$ | $0.363_{\pm0.072}$ | $0.291_{\pm0.018}$ | $1.081_{\pm0.015}$ | $0.072_{\pm0.014}$ | $0.560_{\pm0.016}$ | $0.498_{\pm0.095}$ | $0.341_{\pm0.097}$ |
| TACTIS | C-s | $0.601_{\pm0.004}$ | $0.208_{\pm0.002}$ | $0.634_{\pm0.009}$ | $0.142_{\pm0.015}$ | $1.871_{\pm0.022}$ | $0.254_{\pm0.012}$ | $0.456_{\pm0.003}$ | $0.981_{\pm0.013}$ | $0.276_{\pm0.008}$ |
| | N-s | $0.907_{\pm0.010}$ | $0.320_{\pm0.006}$ | $1.013_{\pm0.010}$ | $0.320_{\pm0.006}$ | $2.309_{\pm0.528}$ | $0.391_{\pm0.016}$ | $1.871_{\pm0.003}$ | $1.170_{\pm0.013}$ | $0.403_{\pm0.013}$ |
| $D^3$VAE | C-s | $0.445_{\pm0.023}$ | $0.266_{\pm0.016}$ | $0.219_{\pm0.009}$ | $0.177_{\pm0.017}$ | $0.312_{\pm0.035}$ | $0.198_{\pm0.020}$ | $0.265_{\pm0.027}$ | $0.257_{\pm0.008}$ | $0.243_{\pm0.035}$ |
| | N-s | $0.662_{\pm0.029}$ | $0.479_{\pm0.027}$ | $0.257_{\pm0.058}$ | $0.263_{\pm0.007}$ | $0.642_{\pm0.070}$ | $0.253_{\pm0.104}$ | $0.926_{\pm0.103}$ | $0.391_{\pm0.038}$ | $0.501_{\pm0.005}$ |
| DPK | C-s | $0.718_{\pm0.011}$ | $0.471_{\pm0.024}$ | $0.556_{\pm0.018}$ | $0.341_{\pm0.062}$ | $0.753_{\pm0.035}$ | $0.784_{\pm0.008}$ | $0.827_{\pm0.007}$ | $0.843_{\pm0.009}$ | $0.728_{\pm0.029}$ |
| | N-s | $1.026_{\pm0.012}$ | $0.725_{\pm0.040}$ | $0.887_{\pm0.023}$ | $0.391_{\pm0.157}$ | $1.130_{\pm0.040}$ | $1.062_{\pm0.005}$ | $1.160_{\pm0.007}$ | $1.165_{\pm0.010}$ | $1.147_{\pm0.056}$ |
| MG-TSD | C-s | $0.430_{\pm0.038}$ | $0.174_{\pm0.009}$ | $0.254_{\pm0.054}$ | $0.129_{\pm0.009}$ | $0.298_{\pm0.025}$ | $0.107_{\pm0.055}$ | $0.528_{\pm0.057}$ | $0.250_{\pm0.073}$ | $0.323_{\pm0.015}$ |
| | N-s | $0.693_{\pm0.083}$ | $0.220_{\pm0.017}$ | $0.394_{\pm0.076}$ | $0.292_{\pm0.046}$ | $0.623_{\pm0.026}$ | $0.155_{\pm0.063}$ | $0.710_{\pm0.058}$ | $0.347_{\pm0.078}$ | $0.638_{\pm0.056}$ |
| KooNPro | C-s | $0.328_{\pm0.037}$ | $0.149_{\pm0.051}$ | $0.165_{\pm0.057}$ | $0.081_{\pm0.020}$ | $0.211_{\pm0.033}$ | $0.057_{\pm0.006}$ | $0.184_{\pm0.022}$ | $0.226_{\pm0.041}$ | $0.204_{\pm0.017}$ |
| | N-s | $0.520_{\pm0.045}$ | $0.224_{\pm0.065}$ | $0.225_{\pm0.028}$ | $0.122_{\pm0.034}$ | $0.313_{\pm0.044}$ | $0.095_{\pm0.012}$ | $0.289_{\pm0.025}$ | $0.330_{\pm0.078}$ | $0.308_{\pm0.030}$ |

Tab.1 presents the $\mathbf{CRPS_{sum}}$ and $\mathbf{NRMSE_{sum}}$ values, averaged over 10 independent runs. In comparison, traditional baseline models (e.g., GP-Copula and Timegrid) have higher $\mathbf{CRPS_{sum}}$ values on most datasets, which indicates their difficulty in handling complex time series data.

Compared to modern deep generative models (e.g., D$^3$VAE and TACTIS), KooNPro achieves a better balance between predictive stability and accuracy. Although MG-TSD shows a slight advantage in $\mathbf{NRMSE_{sum}}$ on certain datasets (e.g., ETTs), its $\mathbf{CRPS_{sum}}$ fluctuates significantly, suggesting an inadequate characterization of distributional uncertainty. KooNPro introduces the variance-aware spectrum to enhance global pattern capturing, avoiding spectrum instability caused by local perturbations, thus achieving higher stability and accuracy in handling complex nonlinear patterns. This demonstrates its effectiveness in high-dimensional time series forecasting.

## 5.3 ABLATION STUDY

In this section, we conduct an ablation study to identify the factors contributing to KooNPro's success in prediction tasks. We first examine the role of the Neural Process (NP), which primarily learns and represents the underlying dynamics across whole time series within $\boldsymbol{S}$. We track the KL divergence between $\boldsymbol{S}_D$ and $\boldsymbol{S}_C$ during the training phase (Fig. 3, upper) on the ETTh1 dataset. It decreases over time, converging with the validation set around the 70th epoch, indicating effective learning of time series dynamics. To assess generalization, we compute the KL divergence between $\boldsymbol{S}_C$ and 150 test time series (Fig. 3, lower). The low divergence suggests $\boldsymbol{S}_C$ reliably captures temporal dynamics, even for previously unseen data.

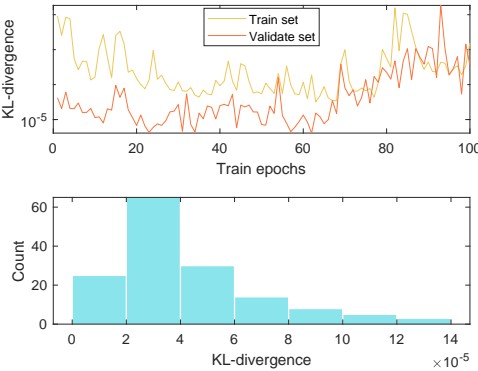

Figure 3: The upper section illustrates the KL-divergence between $\boldsymbol{S}_D$ and $\boldsymbol{S}_C$ during the training phase, with training epochs on the x-axis and an exponential scale on the y-axis. The lower section analyzes 150 test dataset time segments, displaying their KL-divergence with $\boldsymbol{S}_C$, with the x-axis using an exponential scale and the y-axis showing time series counts.

Table 2: Prediction performance when replacing Neural Process (NP) with Attention Neural Process (ANP), Gaussian Process (GP), or removing NP version (denoted as with-ANP, with-GP, and without-NP, respectively).

| Model | Metric | ETTh1 | ETTh2 | ETTm1 | ETTm2 | Solar | Elec. | Traffic | Taxi | Cup |
|---|---|---|---|---|---|---|---|---|---|---|
| KooNPro | C-s | $0.328_{\pm0.037}$ | $0.149_{\pm0.051}$ | $0.165_{\pm0.057}$ | $0.081_{\pm0.020}$ | $0.211_{\pm0.033}$ | $0.057_{\pm0.006}$ | $0.184_{\pm0.022}$ | $0.226_{\pm0.041}$ | $0.204_{\pm0.017}$ |
| | N-s | $0.520_{\pm0.045}$ | $0.224_{\pm0.065}$ | $0.225_{\pm0.028}$ | $0.122_{\pm0.034}$ | $0.313_{\pm0.044}$ | $0.095_{\pm0.012}$ | $0.289_{\pm0.025}$ | $0.330_{\pm0.078}$ | $0.308_{\pm0.030}$ |
| with-ANP | C-s | $0.334_{\pm0.061}$ | $0.194_{\pm0.063}$ | $0.195_{\pm0.029}$ | $0.107_{\pm0.019}$ | $0.201_{\pm0.062}$ | $0.067_{\pm0.022}$ | $0.197_{\pm0.016}$ | $0.213_{\pm0.056}$ | $0.306_{\pm0.008}$ |
| | N-s | $0.541_{\pm0.084}$ | $0.300_{\pm0.093}$ | $0.350_{\pm0.061}$ | $0.182_{\pm0.033}$ | $0.371_{\pm0.052}$ | $0.127_{\pm0.033}$ | $0.308_{\pm0.030}$ | $0.314_{\pm0.081}$ | $0.497_{\pm0.008}$ |
| with-GP | C-s | $0.912_{\pm0.067}$ | $0.613_{\pm0.056}$ | $0.579_{\pm0.023}$ | $0.438_{\pm0.067}$ | $0.304_{\pm0.052}$ | $0.125_{\pm0.033}$ | $0.231_{\pm0.023}$ | $0.349_{\pm0.029}$ | $0.413_{\pm0.159}$ |
| | N-s | $1.210_{\pm0.061}$ | $0.791_{\pm0.058}$ | $0.861_{\pm0.029}$ | $0.622_{\pm0.075}$ | $0.467_{\pm0.067}$ | $0.205_{\pm0.052}$ | $0.388_{\pm0.016}$ | $0.501_{\pm0.033}$ | $0.655_{\pm0.252}$ |
| without-NP | C-s | $0.390_{\pm0.083}$ | $0.218_{\pm0.120}$ | $0.332_{\pm0.068}$ | $0.155_{\pm0.042}$ | $0.341_{\pm0.059}$ | $0.093_{\pm0.027}$ | $0.313_{\pm0.051}$ | $0.252_{\pm0.068}$ | $0.674_{\pm0.224}$ |
| | N-s | $0.609_{\pm0.117}$ | $0.318_{\pm0.169}$ | $0.511_{\pm0.103}$ | $0.243_{\pm0.056}$ | $0.531_{\pm0.065}$ | $0.144_{\pm0.036}$ | $0.590_{\pm0.095}$ | $0.380_{\pm0.092}$ | $1.058_{\pm0.341}$ |

To evaluate the contribution of NP further, we compare different methods for generating $\boldsymbol{S}_C$, including cases without it. For clarity, we label the methods as follows: **With-ANP** (using Attention Neural

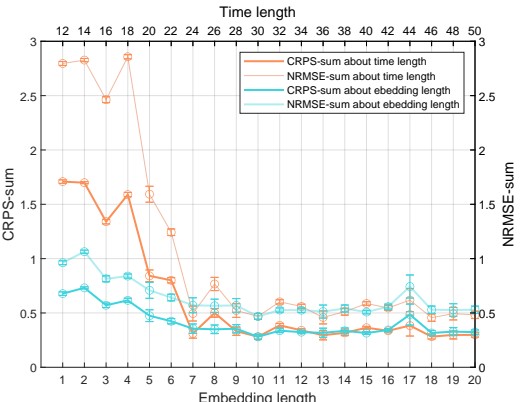

Figure 4: The top axis shows the increase in time length $T$, while the bottom axis indicates the increase in delay-embedded length $k$. The orange and transparent orange curves represent the **CRPS$_{\text{sum}}$** and **NRMSE$_{\text{sum}}$** for increasing $T$. The blue and transparent blue curves correspond to these metrics for increasing $k$. Error bars reflect the standard error from 10 independent retraining and evaluation runs.

Process), **With-GP** (using GP), and **Without-NP** (no NP). Implementation details are provided in Appendix J. As shown in Tab.2, the overall performance of **With-ANP** is comparable to KooNPro, with predictive performance improving as data dimensionality increases. Notably, on the `taxi` dataset, the highest-dimensional dataset, **With-ANP** outperforms KooNPro on both metrics. However, ANP incurs higher computational complexity, increasing from O(n+m) for NP to O(n(n+m)) (Kim et al., 2019). **With-GP** demonstrates significantly degraded performance, even worse than **Without-NP**. This phenomenon may be attributed to the critical influence of prior knowledge in defining the kernel function. Furthermore, the non-learnable parameters may impede $S_D$ from adequately capturing the temporal dynamics governing the entire time series.

Next, we investigate the impact of the hyperparameter time length $T$ and delay-embedded length $k$ on the prediction performance. In Fig.4, we analyze the impact of hyperparameters on the `ETTh1` dataset. The orange and transparent orange curves illustrate changes in **CRPS$_{\text{sum}}$** and **NRMSE$_{\text{sum}}$** as $T$ increases. We observe that KooNPro's prediction accuracy improves significantly with $T$ until $T = 30$, beyond which further increases yield no additional benefits despite higher computational costs. A similar trend is seen when varying $k$, as shown by the blue and transparent blue curves. This suggests that temporal dynamics in time series cannot be fully captured simply by extending the learning time segment. Additional ablation studies across various datasets are presented in Appendix K.

## 5.4 CASE STUDY

The predictive capability of KooNPro arises from its ability to capture the underlying dynamics of time series data effectively, we illustrate this using the `Solar` dataset, which comprises hourly measurements from 137 solar plants located in Alabama state. The dataset exhibits a clear diurnal pattern: values are non-zero from 6:00 to 18:00 and zero at night. In probabilistic prediction, large errors often correspond to high variance, indicating uncertainty. We assess this by analyzing the correlation between prediction variance and accuracy, measured by the mean absolute error (MAE) of the predicted mean relative to the true value. A scatter plot of MAE versus prediction variance (Fig.5, left) shows a correlation of $0.94$ with a significance level near zero, confirming that KooNPro effectively captures temporal dynamics and ensures reliable predictions. The right panel visualizes the ground truth and predictions for the first eight plants over 24 hours. The results demonstrate that KooNPro accurately captures the temporal fluctuations in `Solar` energy generation: The prediction mean approaches zero during the transition from nighttime to daylight and vice versa for each plant. Furthermore, the prediction intervals expand during sunlight and night periods, aligning with the peaks and troughs of `Solar` fluctuations, indicating the reliability of the predictions.

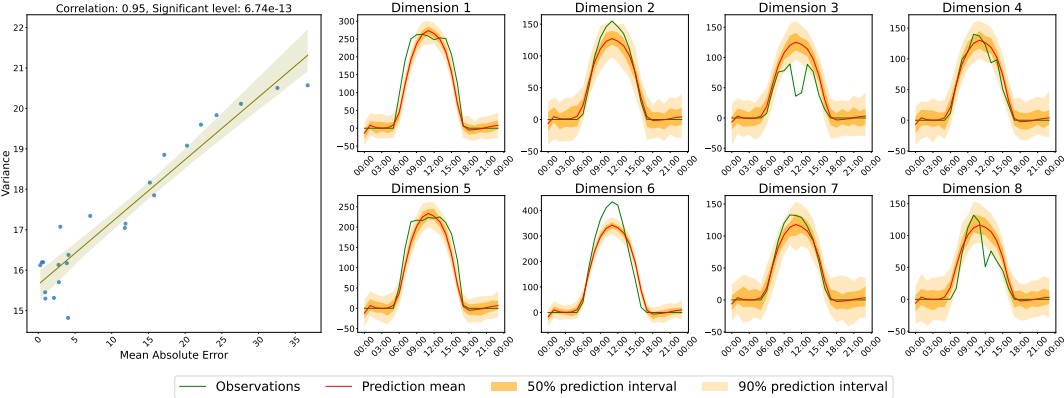

Figure 5: The left panel illustrates the correlation between the MAE of the prediction means relative to the true values and the prediction variance. The title indicates that the correlation achieves $0.95$ and the significant level is $6.74 \times 10^{-13}$. The right panel visualizes the changes in the first eight dimensions of the Solar dataset over 24 hours.

## 6 CONCLUSION

This paper presents KooNPro, a novel probabilistic forecasting model that treats time series as temporal dynamics. KooNPro integrates Neural Process for global temporal dynamics modeling and variance-aware continuous-spectra inspired by pseudo-spectra for local temporal dynamics, enhancing learning capabilities. Extensive experiments on nine real-world datasets demonstrate KooNPro's superior performance compared to state-of-the-art methods. Comprehensive ablation studies explore the origins of KooNPro's predictive power, while visualizations of `Solar` dataset predictions showcase its accuracy and reliability.

## 7 ACKNOWLEDGEMENTS

This work was partially supported by the National Natural Science Foundation of China (No. 12471481, U24A2001), the Science and Technology Commission of Shanghai Municipality (No. 23ZR1403000), and the Open Foundation of Key Laboratory Advanced Manufacturing for Optical Systems, CAS (No. KLMSKF202403).

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

## A   APPENDIX: RELATED WORK

This work engages with three key areas: probabilistic time series prediction, the probabilistic Koopman model, and Neural Process. While each of these fields has received considerable attention in recent literature. Here we talk only about the most related works:

**Probabilistic time series prediction:** Recent developments in probabilistic forecasting for time series integrate deep learning, statistical approaches, and diffusion models. State space models have been significantly enhanced by deep learning, as seen in (Rangapuram et al. (2018b)), (Salinas et al. (2020)) and (Li et al. (2021)), where they effectively capture temporal dynamics and handle missing data. (Feng et al. (2023)) and (Tang & Matteson (2021)) developed attention-based mechanism that enhance long-range dependencies for improved forecast accuracy. Gaussian Process (GP), combined with temporal decomposition, were used by (Yan et al. (2021)) and (Nguyen & Quanz (2021)) to better model uncertainty in multivariate settings. (Salinas et al. (2019)) proposed a Gaussian copula process, and (Ashok et al. (2023)) combines Gaussian copula and the attention mechanism for capturing dependencies in high-dimensional multivariate forecasting. The data distribution-based generative model proposed by (Gouttes et al. (2021)) and the Kalman filter-based approach of (de Bézenac et al. (2020)), blend probabilistic technique for better scalability and robustness. Diffusion models, such as those proposed by (Rasul et al. (2021)), (Li et al. (2022)), and (Fan et al. (2024)), model forecasting as a denoising task, excelling in high-dimensional settings. For faster training and prediction, (Shen & Kwok (2023)) introduced non-autoregressive diffusion model. (Wen et al. (2024)) introduced spatio-temporal diffusion model, extending these methods to spatial dependencies.

**Probabilistic Koopman model:** Over the past two decades, Koopman techniques have garnered substantial attention, with applications spanning analysis (Brunton et al. (2016)), (Takeishi et al. (2017)), (Lusch et al. (2018)), (Azencot et al. (2019)); control Abraham et al. (2017), Korda & Mezić (2018), Kaiser et al. (2021), (Narasingam et al. (2023)); optimization (Dogra & Redman (2020)), (Manojlović et al. (2020)), (Naiman & Azencot (2021)), (Redman et al. (2021)), and forecasting Azencot et al. (2020), Lange et al. (2021), (Wang et al. (2023)), (Liu et al.), (Tayal et al. (2023)). (Brunton et al. (2021)) comprehensively discusses these advances and highlights future research directions. The most closely related work to ours is the probabilistic Koopman model, originally proposed by (Morton et al. (2019)). (Han et al. (2022)) designed a stochastic Koopman neural network for control, in which the latent observables are represented through Gaussian distribution. (Colbrook et al., 2024) integrates the concept of variance into the Koopman framework by introducing variance-pseudo-spectra, thereby ensuring convergence within the model. For time series tasks, (Naiman et al. (2023)) utilized Koopman theory to represent the latent conditional prior dynamics via a linear map, while (Mallen et al. (2024)) introduced a framework enabling probabilistic forecasting for systems with periodically varying uncertainty.

**Neural process:** Neural Process (NP), first proposed by (Garnelo et al. (2018a)) as Conditional Neural Process, bridges neural networks' scalability and Gaussian Process's ability to model uncertainty. (Garnelo et al. (2018b)) provide flexible, probabilistic function approximation, making them efficient and adaptable across tasks, but struggled with long-range dependencies and flexible function distributions. Various models have since emerged to address these limitations. Attentive Neural Process (Kim et al. (2019)) introduced the attention mechanism to handle long-range dependencies better. Convolutional Conditional Neural Process (Gordon et al. (2019b)) adapted convolutional layers to process images and time series data more effectively. Gaussian Neural Process (Bruinsma et al. (2021)) combined NP with Gaussian inference, improving performance in regression tasks by leveraging Gaussian uncertainty. (Lee et al. (2023)) refined uncertainty quantification of NP, providing better posterior estimation. Other works extend NP to meta-learning, such as Meta-Learning Stationary Stochastic Process (Foong et al. (2020)) and Meta-Learning Probabilistic Inference (Gordon et al. (2019a)), both addressing generalization to unseen tasks. Recent works focus on expanding NP applications to complex data structures. Group Equivariant Conditional Neural Process (Kawano et al. (2021)) and Versatile Neural Process (Guo et al. (2023)) improved the handling of symmetry and representation, enhancing NP's capacity to model equivariant and implicit functions. To improve generation ability further, (Nguyen & Grover (2022)) and (Mohseni & Duffield (2024)) combine NP with Transformer and Neural Operator respectively. A comprehensive survey by (Jha et al. (2022)) has organized these developments, exploring their wide applications in uncertainty-aware learning.

# B BACKGROUND

## B.1 KOOPMAN THEORY AND PSEUDO-SPECTRA

The spectrum decomposition of the Koopman operator is a powerful tool for understanding the underlying dynamics. If the operator has a discrete spectrum, it can be expressed in terms of its eigenvalues $\lambda_j$ and eigenfunctions $\phi_j$, where $\mathcal{K}\phi_j = \lambda_j\phi_j$. This decomposition reveals how different modes contribute to the system's evolution. However, many complex systems exhibit continuous-spectra, indicating chaotic or highly irregular behavior. In such cases, the Koopman operator's spectrum is not composed of isolated eigenvalues but rather a continuum, requiring a more nuanced analysis. The operator can be represented through a spectrum measure $\sigma$ and an integral over the continuous-spectra: $\mathcal{K}g(\mathbf{x}) = \int_\sigma e^{i\omega t}dE(\omega)g(\mathbf{x})$, where $E(\omega)$ is a projection-valued measure. This spectrum approach provides a comprehensive framework for capturing both regular and chaotic components of a system's dynamics. While the Koopman operator effectively models dynamics for predictive purposes, it is prone to spectrum pollution, which can impede convergence and stability. Inspired by perturbations in pseudo-spectra and the notion that probability distributions characterize variables through overall dispersion, we propose variance-aware continuous-spectra and model it using Gaussian distribution. This method ensures that eigenvalue perturbations are influenced not only by local disturbances but also by the Gaussian distribution over the dynamics.

## B.2 NEURAL PROCESS

A stochastic process can be viewed as a random function $\mathcal{F} : \mathcal{X} \to \mathcal{Y}$ where inputs can be regarded as indexing the output random variables. With a relaxed use of notation, we employ $p(f)$ in denoting a stochastic process, where $f$ map inputs $x \in \mathcal{X}$ to $y \in \mathcal{Y}$. When fulfilling exchangeability and consistency as stated by Kolmogorov's extension theorem (Oksendal, 2013), Neural Process (NP) cited by Garnelo et al. (2018a) is a stochastic process that describes the predictive distribution over the target set $(\boldsymbol{x}_D, \boldsymbol{y}_D) := (\boldsymbol{x}_i, \boldsymbol{y}_i)_{i \in D}$ given the context set $(\boldsymbol{x}_C, \boldsymbol{y}_C) := (\boldsymbol{x}_i, \boldsymbol{y}_i)_{i \in C}$. Garnelo et al. (2018b) prompts use the distribution of a high-dimensional random vector $\boldsymbol{S}$ to represent $p(f)$ as

$$p(\boldsymbol{y}_D|\boldsymbol{x}_D, \boldsymbol{x}_C, \boldsymbol{y}_C) := \int p(\boldsymbol{y}_D|\boldsymbol{x}_D, \boldsymbol{S})p(\boldsymbol{S}|\boldsymbol{x}_C, \boldsymbol{y}_C)d\boldsymbol{S}. \tag{19}$$

NP consists of two components: an encoder that maps the input-output pairs of the context set $(\boldsymbol{x}_C, \boldsymbol{y}_C)$ to $\boldsymbol{S}$ for representing $p(f)$, and a decoder that combines $\boldsymbol{S}$ with $\boldsymbol{x}_D$ to generate $\boldsymbol{y}_D$. Due to the intractable log-likelihood, NPs adopt amortized variational inference as Kingma & Welling (2014) and maximize the evidence lower bound (ELBO) of the log-likelihood as follows

$$\log(\boldsymbol{y}_D|\boldsymbol{x}_D, \boldsymbol{y}_C, \boldsymbol{x}_C) \geq \mathbb{E}_{\boldsymbol{S} \sim q(\boldsymbol{S}|\boldsymbol{x}_D, \boldsymbol{y}_D)} \left[ \log p(\boldsymbol{y}_D|\boldsymbol{x}_D, \boldsymbol{S}) \right] - D_{KL}\left( q\left(\boldsymbol{S}|\boldsymbol{x}_D, \boldsymbol{y}_D\right) || p\left(\boldsymbol{S}|\boldsymbol{x}_C, \boldsymbol{y}_C\right) \right). \tag{20}$$

# C BENCHMARK DATASETS

For our experiments, we use `ETTs`, `Solar`, `Electricity`, `Traffic`, `Taxi`, and `KDD-cup` open-source datasets, with their properties listed in Tab.C. The dataset can be obtained through the links below.

(i) `ETTs`: https://github.com/zhouhaoyi/ETDataset

(ii) `Solar`: https://www.nrel.gov/grid/solar-power-data.html

(iii) `Electricity`: https://archive.ics.uci.edu/dataset/321/electricityloaddiagrams20112014

(iv) `Traffic`: https://pems.dot.ca.gov

(v) `Taxi`: https://www.nyc.gov/site/tlc/about/tlc-trip-record-data.page

(vi) `KDD-cup`: https://www.kdd.org/kdd2018/kdd-cup

Table 3: Datasets detail

| Name | Frequency | Dimensions | Input length | Prediction length |
|------|-----------|------------|--------------|-------------------|
| ETTh1 | 1 hour | 7 | 10 | 24 |
| ETTh2 | 1 hour | 7 | 10 | 24 |
| ETTm1 | 15 min | 7 | 10 | 24 |
| ETTm2 | 15 min | 7 | 10 | 24 |
| Solar | 1 hour | 137 | 15 | 24 |
| Electricity | 1 hour | 370 | 10 | 24 |
| Traffic | 1 hour | 862 | 10 | 24 |
| Taxi | 30 min | 1214 | 15 | 24 |

## D APPENDIX: EVALUATION METRIC

We consider two metrics: $\mathbf{CRPS_{sum}}$ and $\mathbf{NRMSE_{sum}}$, the first one can describe the predictive distribution, and the second can describe the distance between truth value and prediction mean, more details can be found in Gluonts documentation (Alexandrov et al. (2020)).

$\mathbf{CRPS_{sum}}$: $\mathbf{CRPS}$ is a univariate, strictly proper scoring rule that quantifies the compatibility of a cumulative distribution function $F$ with an observed value $x \in \mathbb{R}$ as:

$$\mathbf{CRPS} = \int_{\mathbb{R}} (\mathbf{F(y)} - \mathbb{I}(\mathbf{x} \leq \mathbf{y}))^{\mathbf{2}} \mathbf{dy}, \tag{21}$$

where $\mathbb{I}(x \leq y)$ denotes the indicator function. The CRPS achieves the minimum value when predictive prediction $F$ same as the data distribution. $\mathbf{CRPS}$ can be extend to $\mathbf{CPRS_{sum}}$ to evaluate multivariate distribution:

$$\mathbf{CRPS_{sum}} = \mathbb{E}_t[\mathbf{CPRS}(F_{sum}^{-1}, \sum_i x_t^i)], \tag{22}$$

where $F_{sum}^{-1}$ is computed by aggregating samples across dimensions and subsequently sorting them to obtain quantiles. A smaller $\mathbf{CRPS_{sum}}$ indicates more accurate predictions.

$\mathbf{NRMSE_{sum}}$: $\mathbf{NRMSE_{sum}}$ is an adaptation of the Root Mean Squared Error (RMSE) that accounts for the scale of the target values. It is defined as follows:

$$\mathbf{NRMSE_{sum}} = \sqrt{\frac{\mathrm{mean}((\hat{\mathbf{Y}} - \mathbf{Y})^{\mathbf{2}})}{\mathrm{mean}(|\mathbf{Y}|)}}, \tag{23}$$

where $\hat{Y}$ represents the predicted time series, and $Y$ represents the true target time series. $\mathbf{NRMSE_{sum}}$ quantifies the average squared difference between predictions and true values across all dimensions, normalized by the mean absolute magnitude of the target values. A smaller $\mathbf{NRMSE_{sum}}$ indicates more accurate predictions.

## E DETAILS ON TRAINING BASELINES

We train baselines by open code which is reported in corresponding papers, and follow the default setting. The code for the baseline methods is obtained from the following sources.

- GP-Copula: https://github.com/mbohlkeschneider/gluon-ts/tree/mvrelease
- Transfomer-MAF:https://github.com/zalandoresearch/pytorch-ts/tree/master/pts/model/transformertempflow
- Timegrad: https://github.com/zalandoresearch/pytorch-ts
- TACTIS: https://github.com/servicenow/tactis
- D$^3$VAE: https://github.com/ramber1836/d3vae

- DPK: https://github.com/AlexTMallen/koopman-forecasting

- MG-TSD: https://github.com/Hundredl/MG-TSD

1.GP-Copula (2019): A method combining Gaussian Process with Copulas to model complex dependencies between variables in multivariate time series, providing uncertainty estimates and capturing nonlinear correlations.

2. Transformer-MAF (2021) Combines Transformers with Masked Autoregressive Flow (MAF) to model long-term dependencies and the conditional probability distribution of time series data effectively.

3. TimeGrad (2021) A diffusion model-based approach for time series forecasting, which progressively generates samples to capture complex dynamics and uncertainty in the data.

4. TACTiS (2023) A probabilistic autoregressive model leveraging Transformers to handle non-stationary time series, focusing on dynamic structures and probabilistic predictions.

5. D3VAE (2023) A deep variational autoencoder (VAE)-based model designed for time series, featuring a dynamic decoder to effectively capture and predict complex temporal structures.

6. DPK (2024) Dynamic Probabilistic Kernel (DPK) models probabilistic dependencies in time series using a dynamic kernel-based approach, balancing flexibility and efficiency for multivariate data.

7. MG-TSD (2024) Multi-Granularity Time Series Decomposition (MG-TSD) decomposes time series into components of varying frequencies or trends, modeling each with a probabilistic framework to capture multi-scale patterns.

## F   APPENDIX: IMPLEMENTATION DETAILS

In Tab.4, we show the hyperparameters of KooNPro include time length $T$, delay-embedded length $k$, layers of the encoder $\phi$, layers of the decoder $\phi^{-1}$, layers of the auxiliary $\kappa$, layers of $\psi$ shown in Eq.3. Note that the choice of $T$ and $k$ is based on the ablation study showcased in Appendix K.

Table 4: Hyperparameters of KooNPro

| Name | $T$ | $k$ | $\phi$ | $\phi^{-1}$ | $\kappa$ | $\psi$ |
|---|---|---|---|---|---|---|
| ETTh1 | 30 | 10 | 4 | 4 | 4 | 2 |
| ETTh2 | 20 | 10 | 4 | 4 | 4 | 2 |
| ETTm1 | 20 | 10 | 4 | 4 | 4 | 2 |
| ETTm1 | 20 | 10 | 4 | 4 | 4 | 2 |
| Solar | 30 | 15 | 8 | 8 | 8 | 3 |
| Electricity | 30 | 10 | 4 | 4 | 4 | 2 |
| Traffic | 20 | 10 | 4 | 4 | 4 | 2 |
| Taxi | 30 | 15 | 6 | 6 | 6 | 3 |
| KDD-cup | 10 | 10 | 4 | 4 | 4 | 2 |

## G   APPENDIX: LONG-TERM PREDICTION

To assess KooNPro's predictive capability in capturing hidden temporal dynamics within time series, we evaluate its performance under extended prediction length, as detailed in Tab.5. According to the results in Tab.6, KooNPro exhibits the least degradation across both metrics compared to baseline methods. This outcome highlights the effectiveness of KooNPro in learning and leveraging the temporal dynamics of time series for accurate prediction.

Table 5: Datasets detail

| Name | Frequency | Dimensions | Context length | Prediction length |
|------|-----------|------------|----------------|-------------------|
| ETTh1 | 1 hour | 7 | 10 | 48 |
| ETTh2 | 1 hour | 7 | 10 | 48 |
| ETTm1 | 15 min | 7 | 10 | 48 |
| ETTm2 | 15 min | 7 | 10 | 48 |
| Solar | 1 hour | 137 | 15 | 48 |
| Electricity | 1 hour | 370 | 10 | 48 |
| Traffic | 1 hour | 862 | 10 | 48 |
| Taxi | 30 min | 1214 | 15 | 48 |
| KDD-cup | 1 hour | 270 | 10 | 72 |

Table 6: Comparison of $\mathbf{CRPS_{sum}}$ (denoted as C-s, smaller is better) and $\mathbf{NRMSE_{sum}}$ (denoted as N-s, smaller is better) across nine real-world datasets. The means and standard errors are based on 10 independent runs of retraining and evaluation. The best performances are highlighted in red and the second are in blue. A block marked with '-' denotes a numerical issue encountered during model training with longer prediction lengths.

| Model | Metric | ETTh1 | ETTh2 | ETTm1 | ETTm2 | Solar | Electricity | Traffic | Taxi | Cup |
|-------|--------|-------|-------|-------|-------|-------|-------------|---------|------|-----|
| GP-Copula | C-s | $0.611_{\pm0.031}$ | $0.381_{\pm0.034}$ | $0.646_{\pm0.056}$ | $0.558_{\pm0.054}$ | $0.465_{\pm0.089}$ | $0.234_{\pm0.047}$ | $0.529_{\pm0.006}$ | $1.007_{\pm0.025}$ | $0.731_{\pm0.025}$ |
| | N-s | $0.909_{\pm0.033}$ | $0.610_{\pm0.062}$ | $0.762_{\pm0.138}$ | $0.844_{\pm0.093}$ | $0.709_{\pm0.085}$ | $0.382_{\pm0.103}$ | $0.713_{\pm0.014}$ | $1.464_{\pm0.057}$ | $1.073_{\pm0.056}$ |
| Trans-MAF | C-s | $1.271_{\pm0.051}$ | $0.507_{\pm0.014}$ | $1.045_{\pm0.085}$ | $0.279_{\pm0.007}$ | - | $0.198_{\pm0.069}$ | $0.596_{\pm0.031}$ | $0.769_{\pm0.046}$ | $0.410_{\pm0.073}$ |
| | N-s | $1.571_{\pm0.144}$ | $0.780_{\pm0.027}$ | $1.797_{\pm0.122}$ | $0.485_{\pm0.127}$ | - | $0.397_{\pm0.127}$ | $0.872_{\pm0.046}$ | $0.936_{\pm0.010}$ | $0.515_{\pm0.083}$ |
| Timegrid | C-s | $0.796_{\pm0.084}$ | $0.477_{\pm0.007}$ | $0.458_{\pm0.059}$ | $0.346_{\pm0.010}$ | $0.886_{\pm0.036}$ | $0.263_{\pm0.028}$ | $0.726_{\pm0.050}$ | $0.791_{\pm0.021}$ | $0.421_{\pm0.059}$ |
| | N-s | $0.953_{\pm0.102}$ | $0.697_{\pm0.009}$ | $0.588_{\pm0.104}$ | $0.455_{\pm0.014}$ | $1.243_{\pm0.056}$ | $0.423_{\pm0.077}$ | $0.932_{\pm0.055}$ | $0.971_{\pm0.195}$ | $0.522_{\pm0.098}$ |
| TACTIS | C-s | $0.752_{\pm0.004}$ | $0.401_{\pm0.001}$ | $1.331_{\pm0.013}$ | $0.261_{\pm0.023}$ | $3.786_{\pm1.708}$ | $0.360_{\pm0.004}$ | $0.552_{\pm0.067}$ | $1.368_{\pm0.014}$ | $0.390_{\pm0.018}$ |
| | N-s | $0.943_{\pm0.004}$ | $0.522_{\pm0.002}$ | $1.853_{\pm0.030}$ | $0.422_{\pm0.023}$ | $5.615_{\pm2.168}$ | $0.498_{\pm0.002}$ | $0.751_{\pm0.023}$ | $1.591_{\pm0.020}$ | $0.505_{\pm0.022}$ |
| D$^3$VAE | C-s | $0.916_{\pm0.036}$ | $0.626_{\pm0.044}$ | $0.598_{\pm0.021}$ | $0.737_{\pm0.064}$ | $0.725_{\pm0.064}$ | $0.408_{\pm0.048}$ | $0.704_{\pm0.069}$ | $0.814_{\pm0.035}$ | - |
| | N-s | $1.265_{\pm0.078}$ | $0.861_{\pm0.041}$ | $0.768_{\pm0.067}$ | $0.951_{\pm0.031}$ | $0.919_{\pm0.141}$ | $0.601_{\pm0.051}$ | $1.126_{\pm0.139}$ | $1.308_{\pm0.164}$ | - |
| DPK | C-s | $0.891_{\pm0.027}$ | $0.744_{\pm0.071}$ | $0.824_{\pm0.040}$ | $0.519_{\pm0.092}$ | $0.938_{\pm0.004}$ | $0.997_{\pm0.012}$ | $1.131_{\pm0.002}$ | $0.969_{\pm0.006}$ | $0.900_{\pm0.015}$ |
| | N-s | $1.262_{\pm0.032}$ | $0.998_{\pm0.138}$ | $1.349_{\pm0.070}$ | $0.592_{\pm0.202}$ | $1.301_{\pm0.004}$ | $1.263_{\pm0.013}$ | $1.478_{\pm0.005}$ | $1.213_{\pm0.005}$ | $1.309_{\pm0.024}$ |
| MG-TSD | C-s | $0.619_{\pm0.056}$ | $0.435_{\pm0.099}$ | $0.371_{\pm0.085}$ | $0.269_{\pm0.005}$ | $1.000_{\pm0.001}$ | $0.174_{\pm0.027}$ | $0.617_{\pm0.045}$ | $0.409_{\pm0.051}$ | $0.590_{\pm0.091}$ |
| | N-s | $0.967_{\pm0.071}$ | $0.627_{\pm0.117}$ | $0.539_{\pm0.109}$ | $0.318_{\pm0.050}$ | $1.610_{\pm1.286}$ | $0.283_{\pm0.035}$ | $0.882_{\pm0.053}$ | $0.621_{\pm0.068}$ | $0.767_{\pm0.073}$ |
| KooNPro | C-s | $0.488_{\pm0.025}$ | $0.376_{\pm0.025}$ | $0.365_{\pm0.018}$ | $0.227_{\pm0.002}$ | $0.417_{\pm0.021}$ | $0.165_{\pm0.017}$ | $0.401_{\pm0.014}$ | $0.396_{\pm0.023}$ | $0.327_{\pm0.014}$ |
| | N-s | $0.746_{\pm0.044}$ | $0.541_{\pm0.035}$ | $0.564_{\pm0.023}$ | $0.343_{\pm0.034}$ | $0.657_{\pm0.040}$ | $0.287_{\pm0.017}$ | $0.715_{\pm0.030}$ | $0.658_{\pm0.042}$ | $0.457_{\pm0.021}$ |

## H APPENDIX: ROBUSTNESS OF PERFORMANCE

To assess the robustness of KooNPro's predictive performance, we evaluate it under varying Signal-to-Noise Ratio (SNR) conditions (20dB, 40dB, and 60dB). During training, KooNPro is trained with ground truth data. In the testing phase, Gaussian noise is added to the input data, and predictions are compared to the ground truth. As shown in Tab.7, the performance degradation with decreasing SNR remains within acceptable limits, demonstrating KooNPro's robust predictive capability across different noise levels.

Table 7: KooNPro represents robust performance across various noisy-add scenarios, with SNR ranging from 20dB to 60dB. Note that KooNPro is trained once on noise-free data and then tested on different noisy-add data.

| Model | Metric | ETTh1 | ETTh2 | ETTm1 | ETTm2 | Solar | Elec. | Traffic | Taxi | Cup |
|---|---|---|---|---|---|---|---|---|---|---|
| KooNPro | C-s | $0.328_{\pm0.037}$ | $0.149_{\pm0.051}$ | $0.165_{\pm0.057}$ | $0.081_{\pm0.020}$ | $0.211_{\pm0.033}$ | $0.057_{\pm0.006}$ | $0.184_{\pm0.022}$ | $0.226_{\pm0.041}$ | $0.204_{\pm0.017}$ |
| | N-s | $0.520_{\pm0.045}$ | $0.224_{\pm0.065}$ | $0.225_{\pm0.028}$ | $0.122_{\pm0.034}$ | $0.313_{\pm0.044}$ | $0.095_{\pm0.012}$ | $0.289_{\pm0.025}$ | $0.330_{\pm0.078}$ | $0.308_{\pm0.030}$ |
| 60dB | C-s | $0.367_{\pm0.023}$ | $0.195_{\pm0.029}$ | $0.183_{\pm0.017}$ | $0.111_{\pm0.012}$ | $0.229_{\pm0.032}$ | $0.074_{\pm0.008}$ | $0.192_{\pm0.023}$ | $0.248_{\pm0.017}$ | $0.217_{\pm0.055}$ |
| | N-s | $0.586_{\pm0.023}$ | $0.284_{\pm0.032}$ | $0.380_{\pm0.021}$ | $0.132_{\pm0.015}$ | $0.323_{\pm0.042}$ | $0.118_{\pm0.014}$ | $0.333_{\pm0.029}$ | $0.337_{\pm0.021}$ | $0.339_{\pm0.105}$ |
| 40dB | C-s | $0.385_{\pm0.028}$ | $0.221_{\pm0.021}$ | $0.271_{\pm0.033}$ | $0.152_{\pm0.001}$ | $0.243_{\pm0.039}$ | $0.192_{\pm0.052}$ | $0.213_{\pm0.019}$ | $0.274_{\pm0.015}$ | $0.229_{\pm0.019}$ |
| | N-s | $0.608_{\pm0.049}$ | $0.317_{\pm0.024}$ | $0.565_{\pm0.048}$ | $0.162_{\pm0.003}$ | $0.359_{\pm0.038}$ | $0.283_{\pm0.061}$ | $0.344_{\pm0.029}$ | $0.380_{\pm0.012}$ | $0.352_{\pm0.031}$ |
| 20dB | C-s | $0.432_{\pm0.021}$ | $0.251_{\pm0.010}$ | $0.311_{\pm0.018}$ | $0.194_{\pm0.017}$ | $0.269_{\pm0.047}$ | $0.287_{\pm0.058}$ | $0.268_{\pm0.016}$ | $0.285_{\pm0.026}$ | $0.249_{\pm0.020}$ |
| | N-s | $0.687_{\pm0.034}$ | $0.331_{\pm0.028}$ | $0.491_{\pm0.026}$ | $0.245_{\pm0.019}$ | $0.385_{\pm0.056}$ | $0.415_{\pm0.049}$ | $0.369_{\pm0.024}$ | $0.384_{\pm0.012}$ | $0.395_{\pm0.036}$ |

## I APPENDIX: THE PARAMETERS OF LEARNED MODELS

The backbone of KooNPro is based on MLPs, with the number of parameters determined by the input data dimensions, as well as the depth and width of the MLP. Tab. 8 compares the memory consumption of KooNPro to baseline methods. Furthermore, Fig.6 illustrates the relationship between memory consumption and predictive performance for each model on the `ETTm2` dataset. The results indicate that KooNPro achieves the best performance with the fewest memory consumption.

Table 8: Comparison of memory consumption for different models' parameters across nine datasets (in MB).

| Model | ETTh1 | ETTh2 | ETTm1 | ETTm2 | Solar | Elec. | Traffic | Taxi | Cup |
|---|---|---|---|---|---|---|---|---|---|
| Trans-MAF | 9.69 | 9.69 | 9.69 | 9.69 | 12.21 | 16.73 | 26.26 | 33.08 | 14.79 |
| Timegrid | 4.22 | 4.22 | 4.22 | 4.22 | 25.93 | 29.32 | 37.83 | 45.06 | 27.82 |
| TACTIS | 7.48 | 7.48 | 7.48 | 7.48 | 7.49 | 7.49 | 7.50 | 7.51 | 7.49 |
| D$^3$VAE | 58.41 | 58.41 | 58.41 | 58.41 | 58.72 | 59.29 | 60.50 | 61.36 | 59.05 |
| DPK | 0.17 | 0.17 | 0.17 | 0.17 | 0.25 | 0.40 | 0.70 | 0.92 | 0.34 |
| MG-TSD | 2.41 | 2.41 | 2.41 | 2.41 | 3.05 | 6.33 | 14.60 | 21.65 | 5.87 |
| KooNPro | 1.73 | 1.73 | 1.73 | 1.73 | 10.23 | 7.18 | 14.42 | 17.50 | 5.60 |

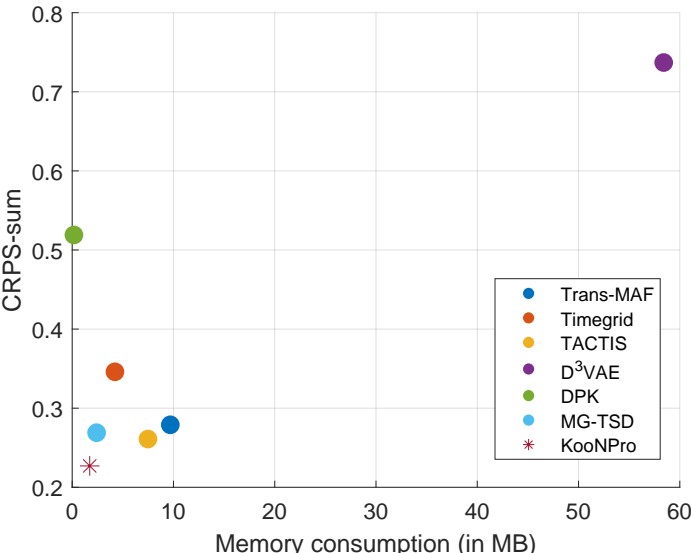

Figure 6: The plot illustrates the relationship between memory consumption and prediction performance of each model on `ETTm2`. The x-axis indicates memory consumption (in MB), and the y-axis represents $\mathbf{CRPS_{sum}}$. Optimal performance is achieved as the marker approaches the bottom-left corner.

## J  IMPLEMENTATION OF NEURAL PROCESS THROUGH DIFFERENT WAYS

This section details the implementation of various methods for generating $\boldsymbol{S}_D$, represented the point-spectra globally and learned by Neural Process. We introduce three KooNPro variants: **Without-NP**, **With-ANP**, and **With-GP**.

- **Without-NP** closely follows the KooNPro procedure. To isolate the impact of $\boldsymbol{S}_D$, the training process adheres to Sec. 4.2, omitting $\boldsymbol{S}$ generated in Sec.4.1. The test procedure is outlined as follows

$$\boldsymbol{z}_{\tilde{T}} = \phi^{-1}(\mathcal{A}^{\tilde{T}-1}(\phi(\boldsymbol{z}_1))). \tag{24}$$

- **With-ANP**, following the Attention Neural Process (ANP) (Kim et al., 2019) and integrating it with our work, employs the attention mechanism to generate $\boldsymbol{S}_D$, which governs the temporal dynamics of the entire time series. In this process, we set the Key $K$ as $\boldsymbol{x}_C$, the Value $V$ as $\boldsymbol{S}_C$, and the Query $Q$ as $\boldsymbol{x}_D$, then compute $\boldsymbol{S}_D$.

- **With-GP** employs the Gaussian process to predict $\boldsymbol{S}_D$ at the target set $\boldsymbol{x}_D$ using context set $\boldsymbol{x}_C$ and their corresponding outputs $\boldsymbol{S}_C$ by modeling a joint Gaussian distribution. The procedure can be summarized as follows

$$\mathbf{K}_C = \mathrm{k}(\boldsymbol{z}_C, \boldsymbol{z}_C) + \sigma_n^2, \tag{25}$$
$$\mathbf{K}_D = \mathrm{k}(\boldsymbol{z}_D, \boldsymbol{z}_D), \tag{26}$$
$$\mathbf{K}_{CD} = \mathrm{k}(\boldsymbol{z}_C, \boldsymbol{z}_D), \tag{27}$$

where $\mathrm{k}$ we choose the radial basis function kernel. Consequently, the distribution of $\boldsymbol{S}_D \sim \mathcal{N}(\mu, \Sigma)$ can be calculated by

$$\mu = K_{CD}^\top K_C^{-1} \boldsymbol{S}_C, \tag{28}$$
$$\Sigma = K_D - K_{CD}^\top K_C^{-1} K_{CD}, \tag{29}$$

where $\mu$ is mean and $\Sigma$ is covariance.

# K   APPENDIX: ABLATION STUDY

We conduct ablation studies on the time length $T$ and delay-embedded length $k$ across several datasets. The results indicate that merely extending the learning time segment is insufficient to reveal the temporal dynamics underlying the time series, as discussed in Sec.5.3. The choices for $T$ and $k$ presented in Tab.4 are guided by the results of this ablation study.

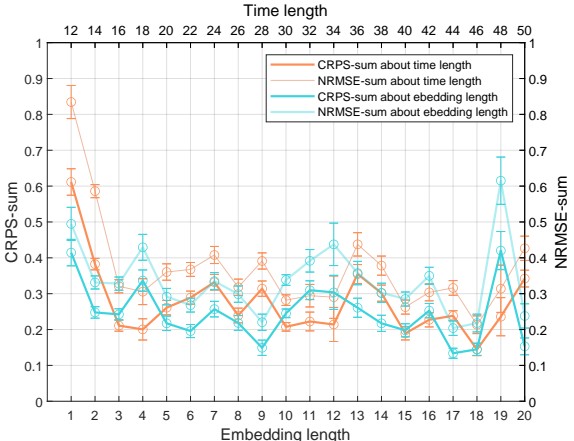

Figure 7: Ablation study on `ETTh2` dataset.

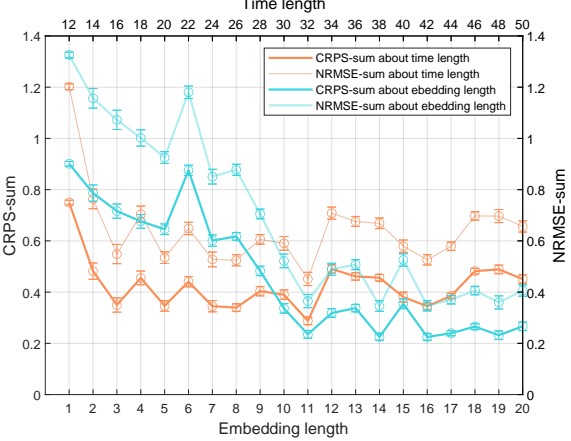

Figure 8: Ablation study on `ETTm1` dataset.

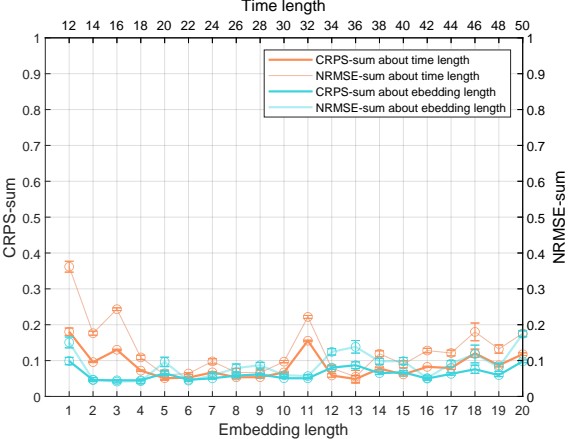

Figure 9: Ablation study on `ETTm2` dataset.

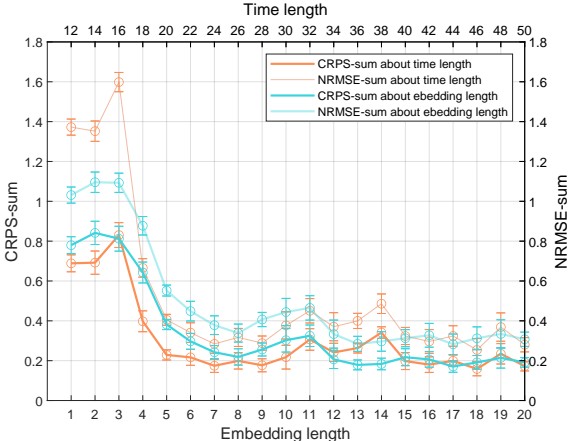

Figure 10: Ablation study on `Solar` dataset.

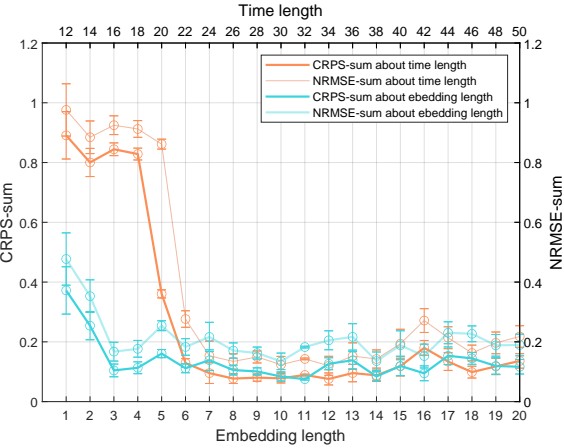

Figure 11: Ablation study on `Electricity` dataset.

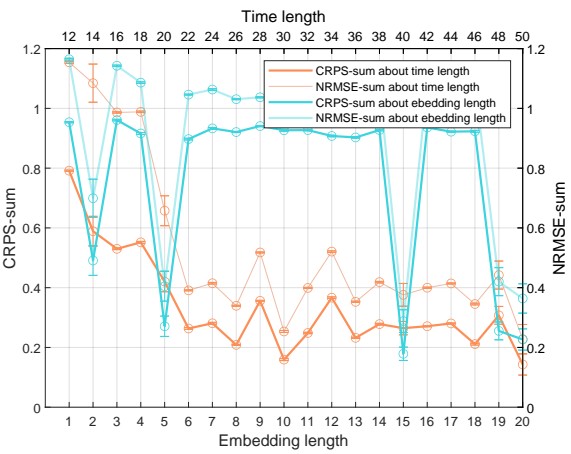

Figure 12: Ablation study on `Taxi` dataset.

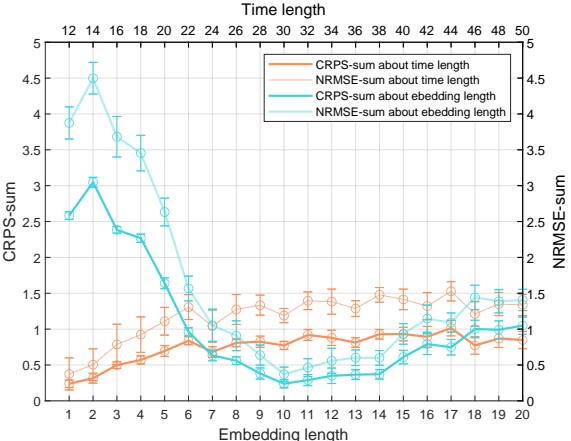

Figure 13: Ablation study on `KDD-cup` dataset.

