# OpenReview forum: "KooNPro: A Variance-Aware Koopman Probabilistic Model Enhanced by Neural Process for Time Series Forecasting"
_ICLR.cc/2025/Conference — ICLR 2025 Poster_

### Official Review · Reviewer_MwLy · 2024-10-31

**Soundness:** 3
**Presentation:** 2
**Contribution:** 3
**Rating:** 6
**Confidence:** 3

**Summary:**

The authors of the paper introduce a novel probabilistic time series forecasting model called KooNPro. KooNPro utilizes a variance-aware continuous spectrum, modeled using Gaussian distributions, to capture complex temporal dynamics and improve stability. By incorporating Neural Processes, the model captures fine dynamics and enhances global modeling capabilities. The authors evaluate KooNPro on nine real-world datasets and find that it consistently outperforms other state-of-the-art models in terms of accuracy and stability.

**Strengths:**

* Deep Koopman models had been used for time series forecasting before but not in conjunction with Neural Processes, hence the proposed model is non-trivial and new.
* The authors enhance their model by a variance-aware continuous spectrum, which to my knowledge has not been done before.

**Weaknesses:**

* Clarity: The authors could improve how they present their work. The math is clunky and too detailed, and can cause the reader to lose attention. For example, in section 4, the authors never refer to Fig.1 more than once (in line 169), so there is a disconnect between the math and the figure. Maybe explain the figure first and detail the model at a high-level, then go in the inner workings and foundations. Also in the caption of Figure 1, consider adding pointers to different parts of the figure to help the reader understand where to look in the Figure for each reference in the caption.

* Better analysis of each component of the model: The authors can provide a more comprehensive and insightful analysis of the Neural Process contribution. They may investigate how different NP architectures (e.g., Attentive Neural Processes, Convolutional Conditional Neural Processes) affect KooNPro's performance.

* Limited empirical discussion on the exact benefits of this model: The authors can better demonstrate For example, conduct experiments to specifically evaluate KooNPro's performance on, say, non-stationary time series or time series tasks with a much larger prediction length etc., comparing it to baselines that do not incorporate NPs. Compared to other models, if there were more convincing and broader experiments demonstrating the benefits of the proposed model, it would lead to wider adoption.

* More empirical evidence for robustness and stability: The paper claims robustness and stability as advantages of the proposed model, but these claims are well empirically validated. For example, the authors can add experiments evaluating the model's performance under different noise levels

* Details on training baselines: Since the baselines were trained for each dataset, the authors should provide details on how the baselines were trained (how hyperparameter selection was done, what protocol was used etc.) in the appendix. It is important to check if the evaluation compares all models in a fair manner.

**Questions:**

* It is not clear what the term "spectral pollution" means in line 143. Consider explaining it for a reader who may not know about Koopman operators.

* Why are all context lengths 10 in Table 4 (Page 15)? Why wasn't the model tried with a different context lengths? This is also concerning as much larger context lengths are used in practice.

---

> ### Author Response · Authors · 2024-11-24
>
> Thank you for your insightful feedback and thought-provoking questions.
> ***
> **[W1]** *Improvememt of Presentation.*
>
> **[A]** We have modified the first paragraphs of Sections 4.1 and 4.2 in the manuscript to strengthen the connection between the details and the overall framework shown in Fig.1. Additionally, links to Fig.1 are added in the first paragraph of Section 4 to clarify the explanation of the framework. These modifications in the manuscript are now,
>
> In Section 4, "... Initially, NP captures the discrete spectrum of dynamics governing the entire time series which is shown by the downward arrows in Fig.1. Additionally, inspired by the concept of pseudospectra, we utilize the probabilistic deep Koopman model to refine these dynamics, obtaining a variance-aware continuous spectrum for prediction  which is demonstrated by the shadowed box in Fig.1"
>
>  In Section 4.1, "The proposed model first identifies the distribution
> of the latent variable  in Eq.1 through an embedding
> which is presented in the left part of Fig.1. The latent variable
> integrates the underlying dynamics present in the time series, which
> allows the model to be more reactive to global features of the time
> series ..."
>
> In Section 4.2, "The probabilistic deep Koopman model shown in the gray block of Fig.1 concentrates on explaining local characteristics of time series, namely intricate temporal dynamics, with a variance-aware continuous spectrum ..."
> ***
> **[W2]** *Better analysis of each component of the model.*
>
> **[A]** We substitute Neural Process with Attention Neural Process and Gaussian Process and test their predictive results in Appendix E.
> ***
> **[W3]** *Performance on non-stationary time series or time series tasks with a much larger prediction length.*
>
> **[A]** The KPSS test (Kwiatkowski–Phillips–Schmidt–Shin test) is a classical statistical method used to determine whether a time series is stationary. We apply the KPSS test to each dataset to assess its stationarity. The results reveal that ETTs are stationary time series, while solar, electricity, traffic, taxi, and cup are non-stationary. The superior performance of KooNPro across these five non-stationary time series demonstrates its effectiveness in handling non-stationary scenarios. It can be found in Appendix G.
>
> We compare each model in the longer prediction length, the details of each dataset, and the results in Appendix F.
> ***
> **[W4]**  *Performance about Robustness.*
>
> **[A]** To evaluate the robustness of KooNPro's predictive performance, we test it under varying Signal-to-Noise Ratio (SNR) conditions (20dB/40dB/60dB). During training, KooNPro is provided with ground truth data. At the testing stage, Gaussian noise is added to the input data, and the predictions are compared against the ground truth. As presented in Tab. 8, the performance degradation with decreasing SNR remains within acceptable limits, demonstrating KooNPro's robust predictive capability across varying noise levels. It can be found in Appendix H.
> ***
> **[W5]** Details on training baselines.
>
> **[A]** We train baselines by open code which is reported in corresponding papers, and follow the default setting. It can be found in Appendix J.
> ***
> **[Q1]** *Definition of spectral pollution.*
>
> **[A]** We apologize for the lack of clarity in explaining the term “spectral pollution.” In the context of Koopman operators, “spectral pollution” refers to the phenomenon where the computed eigenvalues (or spectra) of an approximation to the Koopman operator contain spurious or extraneous components. These unwanted components can arise due to numerical inaccuracies, poor choice of observables, or insufficient resolution in the discretization of the state space. These spurious eigenvalues can distort the true dynamics of the system and make it harder to extract meaningful insights from the system’s behavior.
>
> In simpler terms, when applying techniques like Dynamic Mode Decomposition (DMD) or other spectral methods to approximate the Koopman operator, we expect to recover the true modes and eigenvalues that describe the underlying dynamics of the system. However, in practice, numerical methods can introduce errors that lead to additional, unphysical modes or eigenvalues that do not correspond to the true dynamics. This ``pollution’’ of the spectrum can complicate the interpretation of the system’s modes and can reduce the accuracy of predictions or the stability of the model.
> ***
> **[Q2]** *Long-term Prediction.*
>
> **[A]** The choice of context length is informed by the ablation study results, as presented in Figure 3 and detailed in the Appendix ablation study. The study indicates that increasing the context length does not necessarily improve prediction performance, highlighting the importance of selecting an optimal context length for effective modeling.

---

> > ### Comment · Reviewer_MwLy · 2024-11-28
> >
> > Thank you for the clarifications. I acknowledge the authors' response.
> >
> > The manuscript is much better but my broader perception of the novelty of the paper still remains; I keep my score.

---

> > > ### Author Response · Authors · 2024-12-03
> > >
> > > Thank you for acknowledging the improvements in our manuscript. We understand your concerns about the novelty and have emphasized our main contributions to address this.
> > >
> > > Our innovation lies in modeling **latent dynamics** by extending from discrete point spectra to continuous spectra and incorporating the effects of spectral pollution into a **variance-aware continuous spectral** framework. By capturing dynamic behaviors in the continuous spectrum and modeling its variance, we achieve a more precise description of time series. Additionally, we utilize auxiliary networks to enhance the model's performance.
> > >
> > > Our approach provides a new perspective for time series prediction, revealing the advantages of variance-aware continuous spectral modeling of underlying dynamics in latent space.
> > >
> > > We appreciate your time and constructive feedback, which have been invaluable in refining our work.

---

### Official Review · Reviewer_piba · 2024-11-03

**Soundness:** 3
**Presentation:** 3
**Contribution:** 3
**Rating:** 6
**Confidence:** 2

**Summary:**

The paper presents KooNPro, a novel approach for probabilistic time series forecasting that integrates a variance-aware deep Koopman model with Neural Processes (NPs). By employing a variance-aware continuous spectrum modeled with Gaussian distributions, KooNPro effectively captures complex temporal dynamics with enhanced stability. It leverages NPs to capture global patterns across time series, improving prediction accuracy. Extensive evaluations on nine real-world datasets show that KooNPro outperforms state-of-the-art models, with ablation studies confirming the importance of its components and hyperparameters.

**Strengths:**

1. The proposed variance-aware continuous spectrum modeling address the complex nonlinear temporal dynamics, which is novel and has a solid theoretical foundation.
2. The empirical improvements are significant, outperforming other methods in complex, high-dimensional forecasting tasks.
3. The ablation study demonstrates the effectiveness of Neural Process and help to understanding the role of this key component. The case study shows that KooNPro's predictive capability aligns well with real-world observations, capturing diurnal patterns and demonstrating reliable predictive intervals.

**Weaknesses:**

1. The integration of Neural Processes and the probabilistic deep Koopman model requires a sophisticated training approach, including variational inference for optimizing the ELBO. This could pose challenges in implementation, and could increase the computational complexity.
2. The effectiveness of KooNPro relies heavily on the design of the encoder and decoder networks for the linear space transformation. My concern is that poorly designed architectures could lead to suboptimal representations and thus hinder the model’s ability to learn the underlying temporal dynamics accurately.

**Questions:**

Can you explain more intuitively why complex temporal dynamics can be captured using the proposed probabilistic deep Koopman model? In addition, I'm curious about the validity of the assumptions that formula (6) relies on: "hypothesize the latent space created by $\phi$ possesses linear characteristics", since formula (6) seems to be an important foundation of the subsequent derivations.

---

> ### Author Response · Authors · 2024-11-24
>
> Thank you for these insightful comments.
> ***
> **[W1]** *Questions about challenges in implementation, and the computational complexity.*
>
> **[A]** With advancements in probabilistic deep neural networks, such as variational autoencoders and diffusion models, the calculation of the Evidence Lower Bound (ELBO) has become both stable and efficient. Moreover, KooNPro integrates lightweight MLP architectures, which reduce computational overhead compared to transformer-based and diffusion-based methods.
> ***
> **[W2]** *Questions about the design of the encoder and decoder networks.*
>
> **[A]** That's an insightful question. We have also considered employing different encoder-decoder architectures tailored to specific scenarios. For example, CNNs may be more suitable for image-related tasks, transformers for natural language processing, and GNNs for graph-structured data. Adapting the architecture to the nature of the data ensures better performance and efficiency in diverse applications.
> ***
> **[Q1]** *Questions about the model's ability to capture complex dynamics and the validity of the linearity assumption in formula (6).*
>
> **[A]** The proposed probabilistic deep Koopman model captures complex temporal dynamics via the Koopman operator framework, which allows for a linear representation of nonlinear dynamical systems in a latent space. At its core, the Koopman operator provides a way to track the evolution of observables (functions of the state variables) over time, even in systems with highly nonlinear behavior.
> In our approach, we extend this framework using deep learning techniques to approximate the Koopman operator in a probabilistic manner. The probabilistic aspect allows the model to capture the inherent uncertainty and variability in the system's dynamics. Rather than assuming a rigid, deterministic evolution, the model learns a distribution over possible future states, which helps capture the stochastic nature of complex temporal dynamics that may arise due to noise, unobserved variables, or inherently uncertain processes.
>
> Additionally, the probabilistic formulation provides a way to quantify the uncertainty in the predictions, which is important when dealing with real-world systems where uncertainty is an inherent characteristic. This not only helps in making more accurate forecasts but also provides a framework for analyzing and understanding the system's temporal behavior under varying conditions.
> The assumption in formula (6)—that the latent space possesses linear characteristics—is grounded in the principles of Koopman theory, which links nonlinear dynamical systems to linear representations in an augmented (or extended) space of observables. In Koopman theory, the original nonlinear system is mapped to an infinite-dimensional space where the dynamics are represented by a linear operator acting on the space of observables. This mapping allows us to model the evolution of nonlinear systems using linear techniques, despite the underlying system being nonlinear. More specifically, the "latent space" referred to in our formulation can be understood as a discretization of the observable space, where the dynamics of the system are approximated by a set of chosen observables.
>
> Thus, the validity of the assumption is justified by Koopman theory, which provides a theoretical foundation for understanding the behavior of nonlinear systems in a hidden linear space.

---

> > ### Comment · Reviewer_piba · 2024-11-26
> >
> > Thank you for your response. My concerns have been addressed. I decide to maintain my current score.

---

> ### Author Response · Authors · 2024-12-03
>
> Thank you for your feedback. We appreciate your time and efforts in reviewing our manuscript.

---

### Official Review · Reviewer_xASk · 2024-11-03

**Soundness:** 3
**Presentation:** 2
**Contribution:** 2
**Rating:** 6
**Confidence:** 3

**Summary:**

The authors study the practical problem of probabilistic time series forecasting and address the challenges. They propose KooNPro, which combines two methods, i.e., the Koopman model and Neural Processes.

**Strengths:**

- Addresses important problem of probabilistic time series forecasting
- Interesting idea to use the latent representations from the Neural Processes, typically used in computer vision
- Extensive experiments on 9 time series datasets
- Multivariate forecasting problem is challenging
- Good selection of multivariate probabilistic baselines to compare to including GP-Copula
- Good probabilistic metric CRPS is used in the benchmarking
- KooNPro shows state-of-the-art performance on 8/9 datasets

**Weaknesses:**

- I think more background on the importance of probabilistic forecasting and its use in practical downstream task, e.g., supply chain in the introduction would be helpful
- Overall the writing quality could be improved.
- When discussing LSTMs in the introduction, DeepAR should be cited
- For probabilistic diffusion models, Kollovieh et. al, "Predict, refine, synthesize: Self-guiding diffusion models for probabilistic time series forecasting", NeurIPS 2024 should also be cited.
- There has also been an abundance of foundation models for time series forecasting, e.g., Ansari et. al, "Chronos: Learning the language of time series", 2024 that is not discussed int he introduction.
- DMD is more typically used in the dynamical systems and scientific computing communities
- The motivation for choosing Neural Processes as the probabilistic models rather than more current state-of-the-art probabilistic models, i.e., diffusion models is not clear.
- For application of NPs to spatio-temporal time series (PDEs), see Hansen et. al, "Learning Physical Models that Can Respect Conservation Laws", ICML, 2023
- Some of the background in Section 3 could be moved to an appendix to allow for more novelty of the method presentation and results in the main body.
- The architecture in Figure 1 is also taking a large amount of space.
- I have some concerns on the novelty since the proposed method is just combining two previously proposed approaches.
- May be good to include multivariate time series forecasting problem definition.
- Comparisons to other baselines, e.g., DeepAR (Salinas et. al, 2019) and MQ-CNN (Wen et. al, https://arxiv.org/abs/1711.11053, https://proceedings.mlr.press/v151/park22a/park22a.pdf), MQTransformer (https://arxiv.org/pdf/2009.14799, ) are missing, which could be run on electricity and traffic
- Bold the best in Table 2 or use same convention as Table 1 with color in red but I think bolding would be best for both.
- The method seems off in the case study in Dimension 6 and 8

**Questions:**

1. How can the method generalize past Gaussian distributions to modeling arbitrary distributions?
2. Have the authors tested with the Attentive Neural Process (ANP) Kim et. al, 2019, which shoes better performance than Neural Processes?
3. Could the model also be run with Gaussian Processes? I think a comparison to the simpler GP would be nice to add to motivate the benefit of the NP.
4. What is it about the electricity dataset that TimeGrid shows improved performance?

---

> ### Author Response · Authors · 2024-11-24
>
> Thank you for your thoughtful and detailed feedback. Your comments have greatly helped us improve the clarity and depth of our paper.
> ***
> **[W1-W10 & W13]** *Improvement suggestions.*
>
> **[A]** We have revised the introduction to include more background on probabilistic forecasting and its applications, cited the recommended works, and improved the overall writing quality.
>
> The primary reason for choosing Neural Processes (NPs) as the probabilistic modeling framework is their ability to learn global latent representations through an encoder-decoder structure, which is essential for our variance-aware continuous spectrum modeling. In contrast, diffusion models, while powerful for generating probabilistic forecasts, are not inherently designed to function as encoders that can extract such latent representations.
> ***
> **[W11]** *Novelty.*
>
> **[A]** The motivation for this work stems from capturing different granularities of temporal dynamics. The dot spectrum provides a method to capture the stable characteristics of a time series. To leverage this, we use NP to learn a dot spectrum embedding that governs the entire series. While when facing unstable temporal dynamics, this method may be collapsed. Thus, we employ an auxiliary neural network to manage unstable dynamics, enabling a more detailed representation of temporal dynamics through the continuous spectrum. However, numerical issues often lead to spectrum pollution in the continuous spectrum. To address this, we adopt a variance-aware approach to learn Pseudospectra, offering improved robustness and performance. To some extent, our approach aligns with [1], which analyzes time series at multiple granularities. However, we focus on treating time series from a dynamics-based perspective.
> ***
> **[W12]** *Multivariate time series forecasting problem definition.*
>
> **[A]** A multivariate time series forecasting problem predicts future values of multiple variables from past observations. The input is a time series matrix of shape $[T, N]$ ($T$: time steps, $N$: variables), aiming to forecast $H$ future steps while capturing temporal patterns and variable dependencies.
> ***
> **[W13]** *Comparisons to other baselines.*
>
> **[A]** We compare with DeepAR, and MQ-CNN and show the predictive results in Appendix I. As for the MQ-transformer, it's regret that we haven't found an open code.
> ***
> **[W14]** *The case study in Dimension 6 and 8.*
>
> **[A]** We acknowledge that KooNPro does not perform optimally in capturing the peaks of certain dimensions, likely due to the large range of peak values (spanning from 100+ to 400+). However, the key takeaway from Figure 4 is KooNPro's ability to learn the temporal dynamics of the data. Specifically, when values transition from the bottom to the peak and reverse from the peak to the bottom, KooNPro demonstrates low error and variance. From a data processing perspective, it is characterized by continuous peaks and troughs. Therefore, we believe that the higher variance observed in both the peaks and troughs is reasonable. We believe that modeling diverse probabilistic behaviors with a single model is challenging. This represents an important direction for exploration, and we plan to enhance KooNPro to address this aspect in future work.
> ***
> **[Q1]** *Generalize past Gaussian distributions to modeling arbitrary distributions.*
>
> **[A]** NPs utilize Gaussian distributions as their foundational building block. Their capacity to model arbitrary distributions stems from the flexibility of deep neural networks. This flexibility is further enhanced during the training procedure, which can be summarized as follows:
> - **Individual distributions** are Gaussian, while they can represent different means and variances for different inputs, offering adaptability to data.
> - **The splitting of context and target set**, and the learning to aggregate the context adaptively serve as a prior for the prediction at new data points.
> - **The combination of the latent variable** $S$ **and the observed data** allows the model to capture complex dependencies, effectively mimicking arbitrary distributions.
> ***
> **[Q2 & Q3]** *Test with ANP and GP.*
>
> **[A]** We substitute Neural Process with Attention Neural Process and Gaussian Process and test their predictive results in Appendix E.
> ***
> **[Q4]** *Performance of TimeGrid on the electricity dataset.*
>
> **[A]** The properties of the electricity dataset—high dimensionality, short temporal intervals, heterogeneity, and multivariate dependencies—are well-suited to TimeGrad's design. By combining autoregressive RNNs with diffusion probabilistic models. While in Appendix G we test the predictive performance of each model. The result shows in the longer-term prediction case, our model shows better performance than TimeGrid.
> ***
> Reference
>
> [1] Xinyao Fan, Yueying Wu, Chang Xu, Yuhao Huang, Weiqing Liu, and Jiang Bian. MG-TSD:
> Multi-Granularity Time Series Diffusion Models with Guided Learning Process, March 2024.

---

> > ### Comment · Reviewer_xASk · 2024-11-25
> >
> > I would like to thank the authors for their detailed rebuttal and adding more experiments including the comparison to DeepAR and MQ-CNN and the ablations with the GP and ANP. I will raise my score.

---

> > > ### Author Response · Authors · 2024-11-26
> > >
> > > Thank you for your thoughtful feedback and recognition of our efforts. We are glad that the additional experiments, including the comparisons and ablations, addressed your concerns. Your constructive suggestions have been invaluable in improving our work, and we greatly appreciate your positive evaluation and raised score.

---

### Official Review · Reviewer_VVQE · 2024-11-04

**Soundness:** 3
**Presentation:** 2
**Contribution:** 3
**Rating:** 6
**Confidence:** 3

**Summary:**

A method for multi-variate probabilistic time series forecasting is presented which combines the methods for temporal dynamics modeling using deep Koopman model with Neural Processes. The model consists of an encoder computing the hidden state h_t, a learned Koopman operator 'A' predicting the future hidden states and a decoder used to predict the final values using the hidden state h_t. In addition a neural process is used to model the latent dynamics S of the time series, which is used as an input to the Koopman operator to compute the future states.

**Strengths:**

Strengths
- This paper proposes a new way for probabilistic modeling of multi-variate time series by representing the time series into a low dimensional space using neural processes and using them to model the temporal dynamics of the time series.
- The idea presented in the paper is an intuitive approach for multi-variate forecasting. Modeling the latent dynamics of the time series is an important aspect that is missed by most state-of-the-art modeling approaches. As such, the method presented in this paper is interesting to the time series community.
- The paper presents extensive experimental evaluation and ablation studies demonstrating the effectiveness of the approach.

**Weaknesses:**

Weaknesses
- The description of the model is quite cryptic and not written in a clear manner. See questions below for parts that are unclear.
- While the paper describes everything using mathematical equations, the utility or advantage of such equations/models is unclear. How do each of the components of the model help with a better forecast? (i am looking for a high level motivation for each of the components of the model)
- Several works have introduced Koopman theory into time series modeling, however, there is no clear discussion of existing methods and their comparison with this work. The novelty of the paper is unclear without a clear distinction with existing work.
- Other probabilistic forecasting methods modeling latent factors in time series have not been discussed of compared (for example deep state space models).

**Questions:**

- What is the utility of the neural process latent variable S at a high level? Does it help model the correlations between the different time series? Can it be thought of as an encoding of each time series which also model the inter-correlations between the time series?
- It seems that S is an s-dimensional latent variable meaning that it encodes the whole time series in a low dimensional subspace. Many papers have modeled time series as state space models involving low-dimensional latent decomposition. What is the advantage of the proposed approach over these approaches.
Example papers modeling time series into low dimensional sub-spaces:
  - Wang, Yuyang, et al. "Deep factors for forecasting." International conference on machine learning. PMLR, 2019.
  - Rangapuram, Syama Sundar, et al. "Deep state space models for time series forecasting." Advances in neural information processing systems 31 (2018).
  - Paria, Biswajit, et al. "Hierarchically regularized deep forecasting." arXiv preprint arXiv:2106.07630 (2021).
- What is the utility of the delay embedding dimension k?
- Do context and target sets refer to training and testing time periods? can the context and target sets be interspersed, meaning can this method be used to predict missing values of irregularly sampled time series?

---

> ### Author Response · Authors · 2024-11-24
>
> Thank you for your valuable feedback.
> ***
> **[W1 \& W2]** *Model clarity and component motivation.*
>
> **[A]** We have modified the first paragraphs of Sections 4.1 and 4.2 in the manuscript to clarify the advantage of such a design that considering both the local and global dynamics helps achieve better predictions. The lines at the beginning of Sections 4.1 and 4.2 in the modified manuscript are,
>
> "The proposed model first identifies the distribution of the latent variable $S \in\mathbb{R}_s$ in Eq.1 through an embedding $\boldsymbol \tau $ which is presented in the left part of Fig.1. The latent variable integrates the underlying dynamics present in the time series, which allows the model to be more reactive to global features of the time series ..."
>
> "The probabilistic deep Koopman model shown in the gray block of Fig.1 concentrates on explaining local characteristics of time series, namely intricate temporal dynamics, with a variance-aware continuous spectrum ..."
> ***
> **[W3]** *Comparison with other Koopman-based methods.*
>
> **[A]** We have updated the introduction to explicitly distinguish KooNPro from existing Koopman-based and probabilistic time series models in lines 053 to 061. Specifically, we have detailed the limitations of current methods, such as spectrum pollution and fixed state transition assumptions, and how KooNPro addresses these issues through variance-aware continuous spectrum modeling and integration with Neural Processes.
> ***
> **[W4]** *Other probabilistic forecasting methods.*
>
> **[A]** Actually, in our experiments, all of our comparison methods are multivariate probabilistic forecasting models, many of which explicitly or implicitly model latent factors in time series. We have thoroughly discussed these methods in the Related Work section and demonstrated through experiments that KooNPro outperforms these probabilistic forecasting models across multiple datasets. We have included a detailed introduction of the comparison methods in Appendix I.
>
> Among the methods, we did not directly compare our method with state-space models, but we emphasized that GP-Copula and TimeGrad can be conceptually related to state-space models (SSMs). GP-Copula can capture dependencies in time series in a way similar to how SSMs use latent states to represent system dynamics. Gaussian Processes (GPs) can be used to approximate latent dynamics, akin to the latent state modeling in SSMs. TimeGrad leverages diffusion processes, which conceptually align with continuous-time state-space models. Diffusion models often involve latent representations of system states evolving over time, similar to SSMs.
>
> Deep State Space Models excel at modeling local latent transitions, while KooNPro uses Neural Processes and Koopman operator for global dynamic representation and fine-grained local dynamics to provide a more comprehensive modeling framework.
> ***
> **[Q1]** *The utility of the neural process latent variable $S$ at a high level.*
>
> **[A]** As you mentioned, $S$ is indeed an $s$-dimensional latent variable that encodes the entire time series into a subspace, but it doesn't have to be low-dimensional. At a high level, $S$ captures time-invariant features that represent the global characteristics of the time series. $S$ encapsulates the shared structure and underlying patterns across the entire time series, which at a high level, allows it to effectively model the inter-correlations between different time series.
>
> Therefore, $S$ can indeed be thought of as an encoding of the whole time series, not only summarizing individual series but also capturing the relationships and correlations between them implicitly.
> ***
> **[Q2]** *The advantage of the proposed approach over methods that model time series as state space models.*
>
> **[A]** You are correct that many state space models, such as those cited, involve low-dimensional latent decomposition. Similarly, our approach leverages this concept by extracting global time-invariant features and local time-varying dynamics, with distinct advantages brought by our Koopman-based modeling.
>
> While state space models excel in certain scenarios, their assumptions about linear or fixed dynamics can limit their capacity to handle complex systems. KooNPro's use of Koopman operators for global time-invariant feature extraction, coupled with variance-aware modeling and integration of Neural Processes for local dynamics, offers a more robust and flexible framework for probabilistic time series forecasting.
>
> Methods, such as Deep Factors, Deep State Space Models, and Hierarchically Regularized Forecasting, face limitations in handling nonlinear and non-stationary dynamics due to their reliance on linear assumptions, stationary transitions, or hierarchical priors.
>
> We have added a comparative discussion with these three papers in the introduction section in lines 046-050.

---

> > ### Author Response · Authors · 2024-11-24
> >
> > **[Q3]** *The utility of $k$.*
> >
> > **[A]** The utility of the delay embedding dimension $k$ is grounded in Takens' embedding theorem, which suggests that a sufficient number of time-delayed coordinates can reconstruct the state space of a dynamical system. Specifically, Takens' embedding theorem states that, for a dynamical system with a smooth, differentiable flow, it is possible to reconstruct the system’s attractor in a higher-dimensional space by using time-delayed observations of the system’s state. This reconstruction allows us to analyze and understand the dynamics of the system more effectively, especially in cases where the underlying dynamics are not directly observable.
> > In the context of Dynamic Mode Decomposition (DMD), delay embedding plays a critical role when the spectral complexity of the system exceeds its spatial complexity. DMD is inherently a spectral method that decomposes the observed data into modes corresponding to different temporal frequencies. However, when the system exhibits complex dynamics with temporal interactions that are not immediately obvious in the spatial data, delay embedding helps by embedding the time series into a higher-dimensional space. This extension allows DMD to capture temporal dependencies that might otherwise be overlooked, enhancing its ability to model and predict the behavior of systems with higher-order interactions.
> >
> > Therefore, delay embedding dimension $k$ serves as a critical parameter in handling cases where temporal dynamics are more complex than spatial ones, providing a more robust framework for capturing the system’s underlying modes and improving the accuracy of dynamic predictions.
> > ***
> > **[Q4]** *Context and target sets.*
> >
> > **[A]** The training stage involves both context and target sets, aiming to train an encoder capable of generating temporal dynamics embeddings that capture the governing patterns of the entire time series. During testing, the trained **${S}_{C}$** replaces **${S}_{D}$** used in the training phase. Typically, context and target sets are interspersed during training, as discussed in previous works such as [1]. NPs are predominantly applied in 1D regression and image-implementation tasks, both of which are examples of irregular implementations. While we use Neural Processes (NP) for prediction, we extend their application by integrating time series data with the spectrum of dynamics, rather than employing NP directly. We believe that extending NP to time series analysis is both a logical and meaningful direction for advancing its utility.
> > ***
> > Reference
> >
> > [1] Tuan Anh Le, Hyunjik Kim, Marta Garnelo, Dan Rosenbaum, Jonathan Schwarz, and Yee Whye Teh. Empirical evaluation of neural process objectives. In NeurIPS workshop on Bayesian Deep Learning, volume 4, 2018.

---

### Official Review · Reviewer_ahsz · 2024-11-11

**Soundness:** 3
**Presentation:** 4
**Contribution:** 3
**Rating:** 6
**Confidence:** 3

**Summary:**

- KooNPro is a novel time series forecasting method based on Koopman theory.
- It effectively estimates the process dynamics in a latent space where state transitions are linear.
- It eventually provides context-dependent variance estimates.

**Strengths:**

- Time series tasks, especially probabilistic forecasting, are relevant in many domains and far from being solved.
- The method is presented well and provides many benefits (variance estimation, flexible forecast horizon, nice theory, and practical performance).
- I am not aware of these ideas being presented before.

**Weaknesses:**

1. See Questions below.
2. The definition of target and context sets (ll. 153ff) and the splitting of $z$ (ll. 208ff) were not familiar to me, as they are not common in point forecasting. I would have personally benefitted from more exposition.

#### Minor Comments
- References should not be used as nouns (e.g., ll. 037f, 049f).
- Notation: Eq. (2)/(5) is missing a $p$ on the left-hand side and has odd typesetting of the expectation subscript. Some parens/brackets should be adjusted in height, e.g., in Eq. (3) and (17). In l. 219, $x_C$ and $y_C$ are not bold. L. 473 is the $e$  referring to "scientific notation" common in programming languages (in base-10) or Euler's number?
- One could rename Sec. 5 to Experiment**s**.
- Re. l. 306: Are the conditioning lookbacks of the test and the forecast horizon of validation (etc.) overlapping or not?
- Language: The senctence in l. 821f is odd.
- The combination in Fig. 3 is clever. However, given two separate plots would easily fit side-by-side, it is not worth the extra time needed to decipher the legend/axis labels.
- The abstract in OpenReview should be formatted with Markdown syntax, instead of LaTeX.

**Questions:**

Note: The most important questions are listed first.

1. Regarding Sec. 5.2: The presented results are very impressive. However, a large chunk of the time series forecasting literature performs point estimation instead of full distribution modeling and, therefore, commonly evaluates using MAE/MSE (see, for instance, the overview [here](https://github.com/thuml/Time-Series-Library)). Additionally providing these metrics (possibly in the appendix) would help contextualize the findings in the broader body of work. These methods can also provide probabilistic forecasts, e.g., by learning an output that parameterizes a simple distribution to be sampled from or by methods such as Monte Carlo dropout.
2. Does the method need to be run multiple times to obtain the variance estimation samples?
3. The case study in 5.4 is interesting, yet it makes me question the ability of KooNPro to model the variance of the data truthfully. While I agree that many dimensions are modeled appropriately, the results indicate serious limitations of the method.
	1. Dimensions #3 and #6 show values far outside the 90% interval. Are they within something like the 99% interval since the model learns a possibly appropriate long tail of the distribution? Or does it show some of the method's limitations (which would be fine if acknowledged as such)?
	2. Ll. 446f explains the data characteristics of the Solar dataset. However, the model fails to appropriately model the very low nightly variance of the power production around zero and instead shows a significant variance estimate at, for instance, midnight. Why does that occur?
	3. Combining these two observations, the overall variance estimate appears rather *uniform* and not well-adapted to the dataset.
4. How large are the learned models measured in the number of parameters?
5. Could the method be straightforwardly used to forecast fractional steps into the future?

---

> ### Author Response · Authors · 2024-11-24
>
> We sincerely thank the reviewer for the constructive feedback, which has greatly helped us refine and improve our work.
> ***
> **[W1]** *Formatting, notation, language, section naming, and figure presentation.*
>
> **[A]** We have addressed the issues related to formatting, notation, language, section naming, and figure presentation in the revised manuscript. Additionally, $𝑒$ refers to "scientific notation," commonly used in programming languages. There is no overlap between the training and test sets.
> ***
> **[W2]** *Definition of target and context sets and the splitting of $𝑧$.*
>
> **[A]** This method forms the core idea of the Neural Process (NP) introduced by [1], aiming to enhance generalization when handling unseen data. The context can be viewed as a prior that improves generalization. The splitting of $𝑧$ separates the training dataset into context and target sets, a method that is independent of point or distribution estimation. Previous studies, such as [2] and [3], have employed variants of NP for predicting climate change through point estimation.
> ***
> **[Q1]** *Provide additional metrics, contextualize findings, and compare with probabilistic forecasting approaches.*
>
> **[A]** Utilizing the Monte Carlo method to sample the predictive distribution offers a valuable approach for evaluating predictive performance. To assess predictions comprehensively, we consider the two metrics (CRPS/NRMSE) that strongly relate to MSE/MAE. NRMSE measures the normalized root mean square error between the ground truth and the predicted means. CRPS, a generalization of MAE, quantifies the error between the predictive probability density function (PDF) and a step function at the truth value. In this context, MAE can be viewed as a special case of CRPS, where the PDF is replaced by a step function at the predicted value.
> ***
> **[Q2]**  *Run multiple times.*
>
> **[A]** To ensure performance stability, we conduct multiple runs, averaging results over 10 independent runs. Each table presents the mean and standard deviation of the performance metrics, while Figure.3 includes error bars representing the standard deviation, also estimated from the 10 independent runs.
> ***
> **[Q3]** *Variance modeling accuracy, outliers in dimensions (\#3, \#6), overestimated nightly variance in the Solar dataset, and uniform variance concerns.*
>
> **[A]** We acknowledge that KooNPro does not perform optimally in capturing the peaks of certain dimensions, likely due to the large range of peak values (spanning from 100+ to 400+). However, the key takeaway from Figure 4 is KooNPro's ability to learn the temporal dynamics of the data. Specifically, when values transition from the bottom to the peak and reverse from the peak to the bottom, KooNPro demonstrates low error and variance. From data processing perspective, it is characterized by continuous peaks and troughs. Therefore, we believe that the higher variance observed in both the peaks and troughs is reasonable. We believe that modeling diverse probabilistic behaviors with a single model is challenging. This represents an important direction for exploration, and we plan to enhance KooNPro to address this aspect in future work.
> ***
> **[Q4]** *The number of parameters.*
>
> **[A]** We check the number of parameters of each learned model, and we show the table in Appendix I.
> ***
> **[Q5]** *Forecast fractional steps.*
>
> **[A]** It is a great idea. We evaluate KooNPro on functional Magnetic Resonance Imaging (fMRI) data with a temporal resolution of 0.72 seconds, where it continues to demonstrate exceptional predictive capability. What's more interesting, the prediction variance provides insights into the fluctuations of different brain regions. For instance, the left and right cortexes exhibit similar properties, while both are markedly distinct from the subcortical regions. The visualization of predicted results and variance of different brain areas is shown in Appendix E.
> ***
> Reference
>
> [1] MartaGarnelo, DanRosenbaum, Christopher Maddison, TiagoRamalho, David Saxton, Murray Shanahan, YeeWhyeTeh, DaniloRezende, and S.M.AliEslami. ConditionalNeuralProcesses. In Proceedings of the 35th International Conference on Machine Learning, pp.1704–1713.PMLR, July 2018a.
>
> [2] AndrewFoong, WesselBruinsma, JonathanGordon, YannDubois, JamesRequeima, and Richard Turner.Meta-learning stationary stochastic process prediction with convolutional neural processes. Advances in Neural Information Processing Systems, 33:8284–8295,2020.
>
> [3] Wessel P Bruinsma, Stratis Markou, James Requiema, Andrew YK Foong, Tom R Andersson, Anna
> Vaughan, Anthony Buonomo, J Scott Hosking, and Richard E Turner. Autoregressive conditional
> neural processes. arXiv preprint arXiv:2303.14468, 2023.

---

> > ### Comment · Reviewer_ahsz · 2024-11-24
> > **Concerns adressed convincingly**
> >
> > Thank you for your insightful and clear responses!
> >
> > I have some comments:
> >
> > - **Re [W1]** (formatting): I would advise using notation like $10^3$ for exponenets in the camera-ready version for less ambiguity.
> > - **Re [Q1]**: I was and am aware of the definition and benefits of the selected metrics. Yet, MAE and MSE are commonly used for point forecasts, which are a subset of what KooNPro can provide. Providing them, in addition, would still greatly help judge the performance of KooNPro in the standard benchmark setting.
> > - **Re [Q3]**: I still think the variance is off by a significant amount (the mean is spot on, as you pointed out). However, this is also a fairly challenging task to solve and can indeed be deferred to future work.
> > - **Re [Q5]**: This is a very special feature for forecasting models, and I am happy this has improved the work.

---

> > > ### Author Response · Authors · 2024-11-25
> > >
> > > Thank you for your insightful and constructive feedback! We greatly appreciate your recognition of KooNPro and your valuable suggestions. We will refine the formatting for clarity. Your encouragement of the model's unique features is highly motivating—thank you for helping us improve!

---

### Author Response · Authors · 2024-11-24

We have updated the manuscript and highlighted the changes in green. Several supplementary experiments are reported in the appendix, with the titles also highlighted in green. We offer the catalog as follows:
- **Appendix E**: Forecast fractional steps data.
- **Appendix F**: Substitution of Neural Process by Attention Neural Process and Gaussian Process.
- **Appendix G**: Long-term prediction.
- **Appendix H**: Robustness of performance.
- **Appendix I**: The number of parameters of the learned model.
- **Appendix J**: Comparison with DeepAR/MQ-CNN.

---

### Meta-Review · Area_Chair_Wz4Q · 2024-12-13

**Metareview:**

The paper proposes KooNPro, a novel multivariate time series forecaster that is based on the conjunction of a variance-aware Koopman model with Neural Processes. This yields to improved forecasts with better stability, which is empirically corroborated across several real-world datasets. The reviewers all agree unanimously that the problem is relevant and challenging, they appreciate the intuitiveness of the approach, and find the empirical results compelling. Initial concerns revolved around a necessity to further improve the clarity of the method description, the fact that the method is a bit involved, further experiments that highlight benefits of the proposed method beyond just state-of-the-art performance, and additional beneficial descriptions. In the revision phase, the authors have made several inclusions to the manuscript, e.g. in six appendix sections, which has addressed the majority of the raised comments. As such, the majority of reviewers were satisfied and the AC agrees that the paper is above the threshold for ICLR acceptance.

**Additional Comments On Reviewer Discussion:**

Reviewers have raised several thoughtful comments, have asked for various clarifications and have provided a large amount of pointers for inclusion of additional content to improve the paper. The authors have taken these comments into account carefully and have faithfully addressed the majority of contents through revision and additions to the appendix. In turn, reviewers were generally satisfied with the improvement and ultimately all reviewers agree unanimously that the paper is above the acceptance bar. For some reviewers it remains unclear to the AC why the score indicates marginal acceptance, even tho all concerns seem to have been addressed, but the AC assumes that this may be on accounts of the novelty not being groundbreaking. At the same time reviewers acknowledge the challenge, importance and contributions to the problem at hand, so the AC believes the paper should get accepted.

---

### Decision · Program_Chairs · 2025-01-22

Accept (Poster)